

# A Global End-Member Approach to Derive $a_{\mathrm{CDOM}}(440)$ from Near-Surface Optical Measurements

Stanford B. Hooker†
*NASA Goddard Space Flight Center*
*Ocean Ecology Laboratory/Code 616.2*
*Greenbelt, Maryland 20771*

Atsushi Matsuoka
*Université Laval*
*Takuvik Joint International Laboratory*
*Québec City, Canada G1V 0A6*

Raphael M. Kudela
*University of California Santa Cruz*
*Ocean Sciences Department*
*Santa Cruz, California 95064*

Youhei Yamashita
*Faculty of Environmental Earth Science*
*Hokkaido University*
*Sapporo, Japan 060-0810*

Koji Suzuki§
*Faculty of Environmental Earth Science*
*Hokkaido University*
*Sapporo, Japan 060-0810*

Henry F. Houskeeper
*University of California Santa Cruz*
*Ocean Sciences Department*
*Santa Cruz, California 95064*

---

† *First Corresponding Author:* `stanford.b.hooker@nasa.gov`

§ *Second Corresponding Author:* `kojis@ees.hokudai.ac.jp`

*Declaration of conflict of interest:* None



## Abstract

This study establishes an optical inversion scheme for deriving the absorption coefficient of colored (or chromophoric, depending on the literature) dissolved organic material (CDOM) at the 440 nm wavelength, which can be applied to global water masses with near-equal efficacy. The approach uses a ratio of diffuse attenuation coefficient spectral end members, i.e., a short and long wavelength pair. The global perspective is established by sampling "extremely" clear water plus a generalized extent in turbidity and optical properties that each span three decades of dynamic range. A unique data set was collected in oceanic, coastal, and inland waters (as shallow as 0.6 m) from the North Pacific Ocean, the Arctic Ocean, Hawaii, Japan, Puerto Rico, and the east and west coasts of the United States. The data were partitioned using subjective categorizations to define a validation quality subset of conservative water masses, i.e., the inflow and outflow of properties constrain the range in the gradient of a constituent, plus 15 subcategories of water masses that were not evolving conservatively. The dependence on subcategories was confirmed with an objective methodology based on cluster analysis techniques. The latter defined five distinct classes with validation quality data present in all classes, but which also decreased in percent composition as a function of increasing class number and optical complexity. Four different algorithms based on different validation quality end members were validated with accuracies of 1.2–6.2%, wherein pairs with the largest spectral span were most accurate. Although algorithm accuracy decreased with the inclusion of more subcategories containing non-conservative water masses, changes to the algorithm fit were small when a preponderance of subcategories were included. The high accuracy for all end-member algorithms was the result of data acquisition and data processing improvements, e.g., increased vertical sampling resolution to less than 1 mm and a boundary constraint to mitigate wave focusing effects, respectively. An independent evaluation with a historical database confirmed the consistency of the algorithmic approach and its application to quality assurance, e.g., to flag data outside expected ranges, identify suspect spectra, and objectively determine the in-water extrapolation interval by converging agreement for all applicable end-member algorithms. The legacy data exhibit degraded performance (as 44% uncertainty) due to a lack of high-quality near-surface observations, especially for clear waters wherein wave-focusing effects are problematic. The novel optical approach allows the *in situ* estimation of an in-water constituent in keeping with the accuracy obtained in the laboratory.

*Keywords:* CDOM; absorption coefficient; inversion algorithm; global (oceanic, coastal, and inland) distribution



## 1. Introduction

The colored (or chromophoric, depending on the literature) dissolved organic matter (CDOM) spectral absorption
coefficient, $a_{\mathrm{CDOM}}(\lambda)$, where $\lambda$ is wavelength, has been widely used to investigate terrestrial and oceanic biogeochem-
ical processes, as summarized in the review by Nelson and Siegel (2013). The principal objective of this study is to
evaluate whether a proposed algorithm (Hooker et al. 2013) for deriving $a_{\mathrm{CDOM}}(440)$ from a ratio of diffuse atten-
uation coefficient spectral end members, $K_d(\lambda_1)/K_d(\lambda_2)$, with the shortest wavelength denoted $\lambda_1$ and the longest
denoted $\lambda_2$, can be applied to *global* water masses spanning with equal efficacy. Typically, $a_{\mathrm{CDOM}}(440)$ is determined
in the laboratory using an optical instrument and a water sample obtained *in situ*. In this study the water sample
is collected in temporal close proximity to in-water optical sampling used to derive $K_d(\lambda)$ from profiles sufficiently
close to the water surface that the widest range of wavelengths can be accurately observed.

Although an ability to derive an in-water constituent from optical measurements provides a follow-on opportunity
for synoptic surveys using airborne, and ultimately satellite, remote sensing data, this is not a principal objective of
this study. The reason for de-emphasizing remote sensing possibilities is the airborne and spaceborne instruments
typically available do not provide the spectral range used herein, although a *legacy* pair of wavelengths are considered
below. In addition, the principal parameter used here is not measured directly by a remote sensor. Although $K_d(\lambda)$
can be derived from remote sensing observations for part of the spectrum used herein (Cao et al. (2014), the inversion
is incomplete and the inaccuracies introduced compromise the principal objective of the study, which is to determine
$a_{\mathrm{CDOM}}(\lambda)$ *in situ* with an accuracy similar to what is achieved in the laboratory. Finally, this study includes coastal
and inland ecosystems that are typically too small to be studied using common remote sensing platforms.

The selection of $a_{\mathrm{CDOM}}(440)$ as the parameter of interest for an *in situ* optical inversion algorithm is a consequence
of the relationships between CDOM and the solar illumination of the underwater environment, as follows: a) CDOM
protects microorganisms from harmful ultraviolet (UV) radiation, albeit while reducing photosynthetically available
radiation (Nelson and Siegel 2013); b) CDOM affects the heat content of a water mass, e.g., causing stratification for
brown lakes (Houser 2006); c) CDOM supplies inorganic nutrients, i.e., ammonium, (Bushaw et al. 1996) and can
be a source of labile organic substances (Mopper et al. 1991) through photochemical degradation and mineralization
processes; and d) CDOM is a potentially useful parameter to distinguish and trace water masses in the coastal zone
and open ocean (Nelson et al. 2007 and Tanaka et al. 2016).

The *global* perspective as used herein refers to a generalized concept of sampling a multitude of geographical
areas and watersheds wherein three broad categories are sampled: open ocean, coastal zone (e.g., shelf waters, bays,



estuaries, lagoons, etc.), and inland water bodies (e.g., rivers, lakes, reservoirs, wetlands and marshes, etc.). The 1

near-surface viewpoint is not driven exclusively by the desire to produce data products at all wavelengths. The other 2

reasons for sampling and deriving data products close to the water surface are as follows: 3

1. To establish a technique that can ultimately support remote sensing objectives as the technologies advance, 4

   wherein the spaceborne and airborne approaches obtain data products directly from the sea surface signal; 5

2. To use the same protocols for sampling and deriving data products for all water masses, so the widest dynamic 6

   range in water properties can be considered (the shallowest water depth sampled was 0.6 m); and 7

3. To improve the use of the global solar irradiance observations (obtained with a separate solar reference) in 8

   setting a constraint for the fitting procedures used to derive the in-water data products (Antoine et al. 2013), 9

   the effectiveness of which is related to how close the extrapolation interval is to the surface. 10

The Hooker et al. (2013) $a_{\mathrm{CDOM}}(440)$ algorithm is based on a straightforward principal, as follows: if a water mass 11

is studied optically in a homogeneous near-surface interval of the water column, optical data products can be derived 12

for all wavelengths and the most sensitive parts of the spectral domain are the shortest and longest wavelengths, i.e., 13

the spectral end members. The end members exhibit the greatest range in values as a function of the absorption 14

and scattering processes responsible for the attenuation of light and can be used to describe the evolution of the 15

water mass. Consequently, the end-member signals can be inverted to derive typical constituents as a function of 16

changes in attenuation properties. The greatest sensitivity of the inversion scheme is achieved for a short and long 17

wavelength pair that reliably provide data products while spanning the largest part of the spectral domain. 18

Using a homogeneous near-surface interval to derive all data products ensures the spectral interrelationships 19

coincide with the same water used to determine the in-water constituents by laboratory analysis. The perspectives of 20

natural changes and typical properties are also important, because some water bodies are not automatically assumed 21

to have typical water properties. For example, endorheic lakes are enclosed, so ground seepage and evaporation are 22

the principal outflow mechanisms with evaporation continuously concentrating constituents (e.g., salt, nutrients, and 23

pollution). Over time, a narrowly defined ecosystem evolves to withstand the increasingly extreme conditions, and 24

in some cases, higher-order life ceases to exist (e.g., the Dead Sea only has a significant microbiome community). 25

Endorheic lakes are an end point in the expression of water masses, because the range in the temporal gradient 26

of a constituent, e.g., salt, is somewhat unbounded and the water body does not evolve conservatively due to the 27

significant outflow versus inflow imbalance. For purposes exclusive to this discussion, a conservative water body is 28

defined to have an inflow and outflow of properties that constrains the range in the gradient of a constituent. This 29



natural range is usually established by seasonal factors, although unnatural or atypical stressors can cause significant anomalies, which may or may not be seasonal. Examples of the latter are anthropogenic sources (e.g., pollution or agricultural water diversion) and severe weather (e.g., typhoon-induced bottom resuspension in coastal ecosystems).

Consequently, other water bodies subjected to an unexpected stressor that allows an unbounded evolution in the gradient in a constituent, e.g, long-term drought, are anticipated to not evolve conservatively and the constituents expressed as extreme values as a function of time. Once the unexpected stressor is no longer being applied, the water mass evolves semi-conservatively, wherein the atypical properties are flushed out, and at some point in time the water body reverts to a conservative evolution, i.e., the gradient in the constituent is within an expected or *natural* range.

A global perspective is constructed by assembling overlapping ranges in the natural gradient of the constituent and the optical inversion parameters, as measured in conservative water masses. As long as the assembled dynamic range extends across the anticipated global perspective with a sufficiently dense sampling of clear to turbid water masses, an explicit sampling of every possible global water mass is not deemed necessary. The turbid water endpoint of the dynamic range is somewhat undefined, because of present limitations in obtaining in-water optical measurements in extremely shallow or turbid waters (a case in point, White Lake, is presented below), but the clear water endpoint is defined precisely by the pure water limit, i.e., the absence of any constituents. Consequently, the dynamic range presented herein can only be extended in one direction and these additions will likely involve water masses with necessarily very small volumes, so the global perspective presented here is likely to be only marginally incomplete.

Based on the expected anomalies in the expression of a constituent for a water mass that is not evolving conservatively, it is anticipated that such water masses are not suitable candidates for the validation of a global algorithm established to invert the optical properties of conservative water masses. The degree to which the water masses are not appropriate validation sites is a function of the severity of the stressor creating the non-conservative evolution. For example, a short-term water diversion from an otherwise conservatively evolving lake is expected to create a short-term anomaly in a constituent, whereas a long-term drought is likely to create a time series of increasingly extreme values. Consequently, the objective of the sampling methodology used in this study was to obtain measurements in conservative water masses, as well as water bodies subjected to one or more stressors.

## 2. Methods

The Hooker et al. (2013) study did not include traditional lacustrine water masses. The largest inland water masses in the Hooker et al. (2013) study, Great Bay and Little Bay, are tidal estuaries. Consequently, the new validation



data set includes a large variety of lakes and reservoirs, wherein some were selected precisely because compliance with the original (Hooker et al. 2013) algorithm was not anticipated. These nonconservative water bodies provide an important test of the algorithmic approach, because if they do not appear as outliers with respect to the original algorithm, the principles behind the algorithm are challenged. Another reason nonconservative water bodies are important is they provide an investigative perspective that is helpful for understanding water masses subjected to short- and long-term stressors which result in the water mass evolving atypically, e.g., as a result severe weather and climatic changes (flooding and drought), as well as anthropogenic influences (water diversion and pollution).

To improve the quality of optical measurements obtained in near-surface waters, which is essential for studying shallow ecosystems, a number of methodological advancements were included for this study. Consequently, the methods described herein are distinguished with respect to the original Hooker et al. (2013) research as follows: a) the field sampling involved a significantly enlarged study area, including new water body types (e.g., lakes and reservoirs, more numerous rivers, the marginal ice zone, etc.); and b) the in-water optical instrumentation used more advanced technologies to improve sampling efficiency and data quality, because of the difficulties associated with inland waters (discussed below).

## 2.1 Optical Instrumentation

Observations of apparent optical properties (AOPs) describe a water mass as a function of the solar illumination. A difficulty with obtaining accurate AOPs is minimizing shadows and reflections from the sampling platform the light sensors are deployed on or from, e.g., a vessel, dock, or shoreline. An solution for avoiding platform perturbations is to mount the solar reference above and away from nearby structures (easily achieved on a small boat) and to sample far beyond the in-water perturbation areas with a free-falling profiler. This does not remove the self-shading effect by the profiler itself, which necessitates a correction (Gordon and Ding 1992) for the upwelling radiance, $L_u(\lambda)$. The downward irradiance $E_d(z,\lambda)$ does not require a self-shading correction, e.g., to derive $K_d(\lambda)$, thus, the principal advantages of the approach presented here are it relies on $K_d(\lambda)$ and the $E_d(z,\lambda)$ observations can be made very close to the water surface. The latter allows even highly attenuated wavelengths to be measured.

The optical instrument suite deployed for this study is a handheld, free-falling Compact-Optical Profiling System (C-OPS) that measures both $E_d(z,\lambda)$ and $L_u(z,\lambda)$ while an above-water solar reference simultaneously measures the global solar irradiance $E_d(0^+,\lambda)$, where $z$ denotes depth and $0^+$ is at a height immediately above the water surface (Morrow et al. 2010). This type of instrumentation was deployed by Hooker et al. (2013), except the study





documented herein used advanced radiometers with three gain stages (rather than two) to improve sensitivity. 1

The majority of the profiles herein were obtained with a backplane that included a conductivity sensor for bet- 2
ter water mass characterization, as well as the next-generation Compact-Propulsion Option for Profiling Systems 3
(C-PrOPS), which uses two small digital thrusters to maneuver the backplane (Hooker et al. 2018a) beyond the 4
influence of platform perturbations. Hooker (2014) provides the negative consequences of harder to implement alter- 5
natives. In very shallow waters the profiler was hauled to the surface before the thrusters were used for maneuvering 6
under weak thrust, thereby preventing resuspension of bottom material. 7

A transparent drawing of the next-generation C-OPS with C-PrOPS instrumentation is presented in Fig. 1. 8
The hydrobaric buoyancy chamber has air holes for flooding, and two screws provide access to insert up to three 9
compressible bladders. When the weak thrust holding the profiler at the surface is removed (the profiler is slightly 10
negatively buoyant), the bladders slowly compress and increase the near-surface loitering of the profiler, which results 11
in a vertical sampling resolution (VSR) of 1 cm or less. The VSR is defined as the vertical extent of the extrapolation 12
interval used to derive the data products, e.g., $K_d(\lambda)$, divided by the number of retained data points in the interval. 13

The two digital thrusters are mounted at the same cant angle with respect to the vertical, which directs the weak 14
turbulence from the thrusters downward and below the irradiance instrument, thereby ensuring both light apertures 15
are observing undisturbed water; the opposite occurs if thrust is reversed. When equal thrust is applied, the radiance 16
end cap pushes upwards, the irradiance aperture is simultaneously pulled downwards, and the entire profiler moves 17
away and upwards from the operator holding the sea cable. 18

To steer the backplane like a remotely operated vehicle, differential thrust is applied to the two thrusters (Hooker 19
et al. (2018a) and allows for real-time positioning adjustments, which is a significant advantage in shallow waters, 20
e.g., away from a shoreline or within a wetland. Once the profiler reaches the desired position for obtaining a vertical 21
profile of measurements, or *cast*, outside the perturbation extent of the deployment platform, it is kept in position by 22
maintaining weak forward thrust while holding the sea cable. While at the surface, the pressure transducer measures 23
atmospheric pressure right before a profile commences, which allows a pressure tare of the transducer for every cast, 24
which improves the accuracy of depth measurements (Hooker 2014). 25

When thrust is removed, prior thrust momentum keeps the profiler close to the surface, the thruster-induced bias 26
in the roll axis relaxes, and the profiler descends with stable tilts (the pitch angle is already negligible, because the 27
prior thrust aligns the backplane with almost no pitch angle). There is no righting moment phenomenon when a 28
C-OPS profiler begins sampling, as there is for rocket-shaped profilers (Hooker et al. 2001), so the planar orientation 29





of the C-OPS radiometers is maintained from the start of data acquisition, which significantly improves the VSR.  1

A comparison of C-OPS profiling with and without thrusters verified the improved efficiency to collect high-quality  2
data using thrusters in shallow or deep waters was a factor of two or more (Hooker et al. 2018a). Equally important,  3
either no or minor adjustments to the extrapolation interval used to derive the water-leaving radiance, $L_W(\lambda)$, for  4
each station of replicates were needed, because the efficiency of thruster-assisted profiling minimizes the negative  5
influences of heterogeneity on deriving data products across all wavelengths. The improved temporal efficiency in  6
obtaining replicate casts with thruster-assisted profiling yields a closer temporal matchup between the collection of  7
optical profiles and the water sample.  8

Almost all of the water samples were obtained directly from the surface using a bucket. For some inland waters,  9
the optical profiler was deployed by hand from the shoreline or a dock, because it was not always possible to launch  10
a trailered small boat, e.g., due to the boat ramp being out of service because of extreme conditions (drought and  11
flooding) or disrepair, invasive species regulations, etc. If the vessel could not be launched and a water sample  12
could not be otherwise retrieved from the profiling location, the Compact-Profiler Underway Measurement Pumping  13
System (C-PUMPS) was used (Hooker et al. 2018a). The C-PUMPS accessory provides a $20\,\mathrm{ml\,s^{-1}}$ flow rate from  14
the profiler location and fills a 1 l container in less than 1 min.  15

The efficiencies of thruster-assisted profiling improve the VSR of the optical data. Data retention requires a  16
planar orientation of the light apertures to within 5° of vertical. For inland waters and the coastal zone, the average  17
VSR was 6.0 mm, but for very shallow or turbid waters, the average VSR was 0.9 mm. In comparison, the Hooker  18
et al. (2013) study had a VSR of approximately 10.0 mm. For the open-ocean data, the average VSR was 12.9 mm,  19
but this is primarily because open-ocean profiles were in a more turbulent wave field, so the profiler was ballasted to  20
sink faster and descend to a deeper depth with the available cable. The deep mixed layers of the open ocean mean  21
a slightly coarser vertical resolution is not a limitation.  22

## 2.2 Field Sampling

The Hooker et al. (2013) study area was the Beaufort Sea in proximity to the Mackenzie River outflow, the Gulf  23
of Maine and vicinity, including major portions of its inland watershed (Great Bay and Little Bay, the Piscataqua  24
River, and the Merrimack River) plus minor watershed drainage from the Saco River, the Kennebec River, and a  25
saltwater marsh. A validation data set from observations made in US coastal waters within the southern Mid-Atlantic  26
Bight stretching from the mouths of the Chesapeake Bay to the Delaware Bay were also used. Neither of the two  27



data sets included typical lacustrine water masses. 1

The new sampling area for the study herein included the western United States (i.e., California, Oregon, Washington, Nevada, Utah, and Idaho), Hawaii, Puerto Rico, Japan, the western North Pacific Ocean (e.g., the Kuroshio and Oyashio Currents), the central North Pacific Ocean, the Bering Sea, the Chukchi Sea, and the Beaufort Sea (Fig. 2). The latter is the only water mass that slightly overlapped the Hooker et al. (2013) study. The new data set included sampling in a wide diversity of inland rivers, lakes, and reservoirs, including hypersaline and alkaline lakes. 6

The new field data are divided into the aforementioned three primary categories according to whether or not the sampling station was in the open ocean, coastal zone, or inland waters. The open ocean is defined as offshore waters with a water depth exceeding 200 m. The coastal zone includes near-shore bathymetry of 200 m or less, wherein the adjacent saline waters and shorelands strongly influence each other, and includes islands, bays, deltas, transitional and inter-tidal areas, salt marshes, wetlands, beaches, etc. Inland waters are all other water bodies landward of the coastal low-water line, which are predominantly—but not exclusively—fresh lacustrine and riverine ecosystems. 12

A total of 25 campaigns were conducted with seven different instrument suites including a next-generation *hybridspectral* profiler with fixed-wavelength and hyperspectral detector components (Hooker et al. 2018b). The fixed-wavelength capabilities of all instruments had similar spectral configurations such that all radiometers measured the same nine spectral end members from 320–412 nm and 670–780 nm, plus six other common wavelengths (Table 1). 16

All the optical instruments were calibrated at the same facility by the manufacturer with traceability to the National Institute of Standards and Technology (NIST) as described by Hooker et al. (2018b). The latter is a requirement of the NASA Ocean Optics Protocols (hereafter, the Protocols). The Protocols set the standards for calibration and validation activities (Mueller and Austin 1992), which were revised (Mueller and Austin 1995) and updated over time (Mueller 2000, 2002, and 2003). 21

A total of 318 stations were occupied and 1,230 vertical profiles obtained, which were usually executed as a minimum of three sequential casts at each water-sampling station. The majority of the optical data (733 casts) were obtained with C-PrOPS thruster-assisted profiling (Table 1) and in all three primary categories, whereas optical sampling without thrusters (497 casts) was almost exclusively in the open ocean and the coastal zone wherein heterogeneity was minimal and mixed layers were deep. 26

Duplicate, and sometimes triplicate, water samples were usually collected at each C-PrOPS station. For C-OPS open-ocean campaigns in the Pacific Ocean and the Arctic, which included some coastal zone sampling, a single seawater sample was usually collected. A selected volume of the water sample was filtered (through a 0.22 μm filter) 29



under a gentle vacuum and collected in pre-combusted ($450\,^{\circ}$C for $3\,$h) borosilicate glass vials with acid-cleaned $\quad$ 1

teflon-lined caps or clean glass bottles. The filtrate was either stored immediately at approximately $-30\,^{\circ}$C until $\quad$ 2

subsequent laboratory analysis at a shore facility or analyzed onboard ship within a few hours after sampling using $\quad$ 3

a liquid waveguide system, UltraPath (World Precision Instruments, Inc.) $\quad$ 4

$\quad$ The surface water sample was obtained as quickly as possible after three optical casts were performed. In $\quad$ 5

some cases, when the heterogeneity or turbulence of the water mass was considered to be excessive, an additional $\quad$ 6

three optical casts were executed immediately after the water sample was collected. The determination of excessive $\quad$ 7

conditions was based on the stability achieved in optical variables (e.g., the average vertical tilt in the upper $2\,$m of $\quad$ 8

the water column, the depth of the 10% light level, etc.) during data acquisition for the first three casts. $\quad$ 9

## 2.3 Optical Data Processing

$\quad$ Like all optical data products discussed herein, $K_d(\lambda)$ values were estimated in a near-surface interval of the $\quad$ 10

water column with homogeneous properties within which all data products were derived for all wavelengths. The $\quad$ 11

homogeneity of the extrapolation interval was confirmed with physical data, e.g., temperature and conductivity $\quad$ 12

(as available), and analysis of the linearity of spectral attenuation within the layer. The significance of acquiring $\quad$ 13

high-quality optical data close to the sea surface is expressed directly in the processing scheme used to derive the $\quad$ 14

data products. The processor used here is based on a well-established methodology (Smith and Baker 1984) that $\quad$ 15

Hooker et al. (2001) showed is capable of agreement at the 1% level within an international round robin, when the $\quad$ 16

processing options are as similar as possible. This level of achievement requires strict adherence to the Protocols $\quad$ 17

for both data acquisition and processing. Summary details of the data acquisition and processing capabilities are $\quad$ 18

provided in Antoine et al. (2013), Hooker (2014), and Hooker et al. (2018a, 2018b, and 2018c). $\quad$ 19

$\quad$ The Protocols are detailed, so only a brief overview for obtaining data products from vertical profiles of the light $\quad$ 20

field are presented. In-water radiometric parameters in physical units are normalized with respect to a separate $\quad$ 21

radiometer simultaneously measuring the global solar irradiance, $E_d(0^+,\lambda,t)$, with $t$ explicitly expressing the time $\quad$ 22

dependence. Data are only acquired if $E_d(0^+,\lambda,t)$ changes are slowly varying, e.g., changes due to the solar transit, $\quad$ 23

to ensure the correction does not introduce unwanted variance. For simplicity, the temporal dependence and data $\quad$ 24

wherein the vertical tilt exceeds $5^{\circ}$ are omitted in what follows. $\quad$ 25

$\quad$ After solar normalization and tilt filtering, a near-surface portion of $E_d(z,\lambda)$ centered at $z_0$ and having ho- $\quad$ 26

mogeneous properties (verified with temperature, salinity, attenuation, and optical parameters) extending from $\quad$ 27





$z_1 = z_0 + \Delta z$ and $z_2 = z_0 - \Delta z$ is established separately for the blue-green and red wavelengths; the UV and near

infrared (NIR) wavelengths are included in the blue-green and red intervals, respectively. Both intervals begin at the

same shallowest depth, but the blue-green interval is allowed to extend deeper if the resulting linearity in $\ln\left[L_u(z,\lambda)\right]$,

as determined statistically, is thereby improved (this only occurs in oligotrophic, optically simple, water masses with

deep mixed layers). The negative value of the regression slope yields $K_d(\lambda)$, which is used to extrapolate the fitted

portion of the $E_d$ profile through the near-surface layer to null depth, $z = 0^-$.

A principal benefit of profile data with a high VSR is that the aliasing caused by near-surface fluctuations in

the light field from surface waves, so-called *wave-focusing* effects (Zaneveld et al. 2001), can be significantly reduced

during data processing. A value just below the surface (at null depth $z = 0^-$) can be compared to that measured

contemporaneously above the surface (at $z = 0^+$) with a separate solar reference using

$$E_d(0^-,\lambda) \;=\; 0.97\, E_d(0^+,\lambda), \tag{1}$$

where the constant 0.97 represents the applicable air-sea transmittance, Fresnel reflectances, and the irradiance

reflectance, and is determined to an accuracy better than 1% for solar elevations above 30° and low-to-moderate

wind speeds. The distribution of light measurements at any depth $z$ influenced by wave focusing effects do not follow

a Gaussian distribution, especially $E_d$ during clear-sky conditions, wherein the amplitude of the brightened signals

significantly exceed the companion darkened signals. Consequently, arithmetic averaging is not appropriate and the

linear fitting of $E_d$ in a near-surface layer is poorly constrained, especially if the number of samples is small.

The appropriateness of the $E_d$ extrapolation interval, initially established by $z_1$ and $z_2$, is evaluated by determining

if (1) is satisfied to within the 2.3–2.7% uncertainty ($k = 2$ coverage factor) of the optical calibrations (Hooker et al.

2018b); if not, $z_1$ and $z_2$ are redetermined—while keeping the selected depths within the shallowest homogeneous layer

possible—until the disagreement is minimized (usually to within 5% to include some inevitable variance from natural

processes to the calibration uncertainty). In this procedure, selection of the near-surface extrapolation interval uses

a boundary condition or constraint (Antoine et al. 2013), wherein the central tendency of the distribution of data

within the extrapolation interval, which are typically subjected to wave focusing effects, satisfies (1).

The linear decay of all light parameters in the chosen near-surface layer are then evaluated, and if linearity is

acceptable, the entire process is repeated on a cast-by-cast basis. Subsurface quantities at null depth are obtained

from the slope and intercept given by the least-squares linear regression versus $z$ within the extrapolation interval

specified by $z_1$ and $z_2$. A secondary benefit of profile data with a high VSR is that the extrapolation interval can have



a restricted vertical extent, but still have sufficient data to satisfy (1) and produce data products at all wavelengths. <sub>1</sub>

a restricted vertical extent, but still have sufficient data to satisfy (1) and produce data products at all wavelengths.

This is an important advantage in optically complex water masses, which are usually turbid and shallow.

## 2.4 Water Sample Processing

For the Pacific Ocean samples and approximately half of the Arctic samples, the absorption spectrum of CDOM was determined using a spectrophotometer (Shimadzu UV-1800) according to Yamashita et al. (2013). Briefly, after the water sample was thawed and reached room temperature, the spectral absorbance was measured from 200–800 nm at 0.5 nm intervals with a 10 cm quartz-windowed cell. Absorbance spectra of a blank (Milli-Q water) and samples were obtained against air, and a blank spectrum was subtracted from each sample spectrum. The blank-corrected absorbance spectrum was baseline-corrected by subtracting average values ranging from 590–600 nm (Yamashita and Tanoue 2009), and then converted to the absorption coefficient. A single absorbance analysis was generally carried out for the open ocean samples. An average accuracy for the analysis determined with replicates was 2.5%.

For the other half of the Arctic samples, CDOM absorption coefficients were determined using an UltraPath liquid waveguide (Matsuoka et al. 2012 and 2017). Briefly, CDOM absorbance for a filtrate (less than 0.2 μm) was measured relative to a reference water within a few hours after sampling. The reference water was prepared in advance using pre-combusted pure salt (450 °C for 4 h) with Milli-Q water to adjust salinity within ±2 between a sample and a reference to correct for the refractive index effect. A 2 m optical path was used for all waters except some coastal sites wherein a 0.1 m path was used (Matsuoka et al. 2012 and 2014). Absorbance was converted into absorption coefficients by including the optical path length. The detection limit was within $0.001\,m^{-1}$ (Matsuoka et al. 2017).

For the western US coastal and inland waters, CDOM was quantified as the absorption coefficient at 440 nm of the colored dissolved materials that pass through a 0.2 μm syringe filter (Whatman GD/X) and measured on either a Cary Varian 50 spectrophotometer using a 10 cm quartz cell or an UltraPath liquid waveguide spectrometer with 2 m path length. The syringe filter was rinsed with sample prior to collection, with the sample stored in an amber, acid-washed and combusted (450 °C for 4 h) glass vial with Teflon septa, and kept in the dark at 4 °C until analysis. Absorption spectra of the filtered samples were measured using ultrapure water from a Millipore Milli-Q A10 pure water system with UV to reduce total organic carbon to less than 10 ppb. The absorption coefficient (calculated as absorbance divided by path length, multiplied by 2.303 to convert to natural log units) at 440 nm to represent CDOM abundance was estimated using the Single Exponential Model (SEM) for absorption from 300–700 nm as described by Twardowski et al. (2004).



## 2.5 Data Subcategories

A successful validation exercise for an algorithm requires an assessment as to whether or not a sampled water mass violates any of the assumptions made to establish the algorithm. For this study, all water masses wherein the evolution of $a_{\mathrm{CDOM}}(440)$ or $K_d(\lambda)$ was hypothesized to be conservative, i.e., within the likely range in the gradient of such properties because no stressors to challenge that perspective were evident, are considered validation quality and categorized using the three primary sampling categories: open ocean, coastal zone, or inland waters. Those water masses that were considered to be possibly violative of the algorithmic approach, because one or more stressors challenging the conservative evolution perspective were evident, are further subcategorized, as follows:

1. Waters closer to an ice field contain anomalous properties from meltwater, which freshens the neighboring water body and can result in additional particles or compounds not usually found in the parent water mass.

2. Waters farther from an ice field, but within proximity, containing lesser amounts of anomalous melt water.

3. Resuspension occurs naturally when a sufficient flow (e.g., an ebb or flood tide) or turbulent wave field (e.g., created by sufficiently strong winds) interacts with shallow bottom sediment to create concentrations of constituents that would otherwise not be present; it occurs unnaturally when a boat propeller (or other mechanical device) churns up shallow bottom sediment (e.g., in a harbor, marina, or navigation channel).

4. A refilled lake experiences a rapid inflow of alluvium (e.g., gravel, sand, silt, and clay) from riverbeds and eroding banks, plus floating and partially submerged debris, that can also resuspend bottom sediment. If the refilled lake is a controlled reservoir and exceeds the normally maintained fill level, new lake bottom is added, which can be a source for additional, perhaps atypical, water constituents in terms of type or concentration.

5. A drought-stricken lake has a longer residence time (the amount of time for the time-elapsed outflow to equal the lake volume) than normal, because once the water level remains below the overflow elevation, evaporation and ground seepage are the primary outflows. Increased residence time can concentrate in-water constituents, plus dried and exposed bottom material can be resuspended into the shrinking lake during wind or rain events.

6. A harbor (or marina) is a docking facility, usually in shallow water, for vessels of varying sizes. Such facilities can be a source of pollutants and bottom resuspension, and typically include structures (breakwaters, jetties, piers, etc.) for shelter from severe weather, which can alter residence times by restricting water exchange.

7. A harmful algal bloom (HAB) is a toxic or nuisance algae that produces harmful effects on natural resources when present in high concentrations. HABs are usually influenced by chemical, physical, and biological factors.





8. A wetland (plus marsh or mangrove) filters dissolved and suspended water constituents (e.g., from tidal cycles, weather events, etc.) through settling and plant consumption, but might not completely remove them.

9. A polluted water mass is contaminated from an anthropogenic source that alters the natural water properties.

10. An alkaline (or soda) lake has limited biodiversity due to an elevated pH of 9–12 with high carbonate and complex salt concentrations affecting the solubility and toxicity of chemicals and heavy metals (Grant 2006).

11. A hypersaline lake contains high concentrations of sodium chloride (or other salts) surpassing seawater, which limits biodiversity to organisms tolerating high saline levels (e.g., Mono Lake had a salinity of about 50).

12. A river mouth is where significant amounts of alluvium are deposited into a larger water body (e.g., a delta).

13. An atypical algal bloom is based on local reports evaluated with respect to typical conditions, both in time and space, and may involve weather effects (e.g., wind) concentrating algae through advective processes.

14. An invasive species is an introduced plant, fungus, or animal that is not native to a water body and is anticipated to alter the heretofore established properties and perhaps with a significantly negative outcome (e.g., damage to the environment, economy, or health of organisms, including humans).

15. A parent water mass modifier is a localized alteration of water properties, e.g., a creek inflow into a lake, and demonstrates the sensitivity of the methods used herein to distinguish small changes.

The above 15 subcategories plus the original validation quality category results in 16 possible categorizations. If a sample was considered applicable to more than one subcategory, e.g., a marsh area can experience resuspension from tidal currents, a single (presumed dominant) subcategory was selected based on observations during sampling.

The proximity to ice (farther and closer) subcategories are based on the relative position of safely operating a small vessel in and around an ice field. Sampling was usually as close to the ice as possible, and then as far from the ice within line of sight of the larger ship the small boat was launched from. In comparison, categorizing a refilled lake is a straightforward comparison of the water level datum available from local authorities with respect to the outflow elevation and historical norms. All refilled lakes were at 100% capacity or more, e.g., Washoe Lake was overfilled.

The categorization of bottom resuspension is primarily based on visual evidence, wherein resuspended particles are visible and produce a significant change in water color (e.g., Akkeshi Bay the day after the passage of typhoon Vongfong). Although a subset of sampling obtained in harbors could be classified as resuspension stations, a harbor (or marina) is identified based on local identification of such facilities. Similarly, wetlands (plus marshes and mangroves) are identified based on navigation charts, maps, and local descriptions. Hypersaline (endorheic) lakes



are similarly categorized by state and local authorities (e.g., Mono Lake, Great Salt Lake, and Salton Sea), as are

alkaline lakes (e.g., Mono Lake, with dual classification, plus Borax Lake and Soda Lake).

Categorizing drought-stricken lakes relies on local authorities reporting lake elevations and inflow water volumes

with respect to historical norms. Examples from 2015 are as follows: a) Shasta Lake water storage was 56% below

normal storage; b) Lake Almanor was 118.5 ft (36.1 m) below normal elevation; c) the Truckee River flow into Pyramid

Lake (Nevada) was near historical lows and was dry for three days prior to sampling; and d) Eagle Lake had a water

level of 5,091.5 ft (1,551.9 m), which was within 0.5 ft (0.2 m) of the lowest level recorded in 1935.

The categorization of a river mouth is a combination of the geographical location (e.g., the Columbia River) and

evidence of the presence of the water mass the river flows into (e.g., salt water intrusion from the near-shore bay).

The inflow of smaller rivers, streams, and creeks into a larger water body (e.g., Ward Creek flowing into Lake Tahoe)

are not classified as river mouths, but rather as parent water mass modification from a creek inflow. The creek

designation is to ensure the understanding that the inflow volume is small, but the expectation is anomalous water

properties are nonetheless discernible, because of the enhanced sensitivity (e.g., VSR) of the methods used.

The categorization of a water mass subjected to pollution, an atypical bloom, HAB, or an invasive species relied

principally on eutrophic chlorophyll concentrations plus historical reporting or on-site local representatives. The

latter are frequently present at boat ramps to oversee measures to mitigate health concerns or prevent the spread of

invasive species, which is presently a significant and escalating problem throughout the western United States.

## 3. Results

All data not categorized as one of the 15 subcategories provided above (Sect. 2.5) are retained in the open ocean,

coastal zone, and inland waters primary categories to yield the following number of validation quality observations,

respectively: 190, 223, and 196. This new filtered data set of 609 observations is reasonably balanced, in the sense

that each of the primary categories contains approximately 200 observations. It is these data that are used to initially

evaluate the global applicability of the original Hooker et al. (2013) $a_{\mathrm{CDOM}}(440)$ algorithm.

The comparison of the new validation quality data (i.e., data not part of the 15 subcategories) with respect to the

original algorithm is presented in Fig. 3, wherein the location names of a subset of open ocean (blue), coastal zone

(green), and inland waters (red) are explicitly identified as a function of the approximately three decades of dynamic

range in both axes. The most important distinctions of the new data set with respect to Hooker et al. (2013) are

as follows: a) the addition of lacustrine water bodies (including Lake Shikotsu in Hokkaido, Japan), which almost





span the entire three decades of dynamic range (e.g., Crater Lake to Pinto Lake); b) the expansion of the dynamic 1

range to include clearer and more turbid water masses, which also means deeper and shallower waters; and c) a more 2

global sampling (e.g., western US, Hawaii, Puerto Rico, Japan, the Kuril Islands, Chukchi Sea, Bering Sea, etc.). 3

The new validation quality data significantly adhere to the original algorithm, as evidenced by how the red, 4

green, and blue circles in Fig. 3 are well contained within the approximately ±15% gray boundaries that denote the 5

dispersion in the original algorithm. The category that spans the largest percentage of the dynamic range for both 6

axes is the inland waters data, although the coastal zone is somewhat similar because of the clear Hawaiian coastal 7

waters that were sampled. The open ocean category has the smallest dynamic range, but this does not diminish the 8

importance of this category because it represents the greatest surface area and volume of water on the planet. 9

The new data values, $V_n$, are compared to the corresponding algorithm value, with the latter being the reference 10

values, $V_r$, in the comparison calculation. The relative percent difference (RPD) between the new data and the 11

algorithm is computed as RPD $= 100\,(V_n - V_r)/Vr$, and is expressed as a percent. The average RPD for all the 12

new data is 0.02%, i.e., the new data show a negligible bias with respect to the original algorithm. The absolute 13

percent difference (APD), which provides an estimate of the dispersion of differences between the new data and the 14

algorithm, is the absolute value of the RPD. The average APD value for all the new data is 3.86%, i.e., the new 15

validation quality data are usually to within 5% of the original algorithm (as visually confirmed by Fig. 3). 16

## 3.1 Drought-Stricken, Alkaline, Hypersaline, and Refilled Lakes

The new lacustrine data are presented in Fig. 4. Data from the hypersaline and alkaline (endorheic) lakes do not 17

conform with the algorithm. Drought-stricken lakes exhibit a wider range of departure, with the most significant 18

occurring for the most depleted water bodies, e.g., Lake Almanor and Shasta Lake. Endorheic drought-stricken lakes, 19

e.g., Eagle Lake and Pyramid Lake, are the most extreme. Refilled lakes also do not conform with the algorithm, 20

and refilled drought-stricken lakes exhibit an increase in CDOM and turbidity, e.g., Shasta Lake and Pyramid Lake. 21

The three refilled Shasta Lake samplings in Fig. 4 were conducted in different locations. The small differences 22

in the data are associated with local changes to the parent water mass caused by the inflow of a creek, as well as a 23

large floating and partially submerged debris field blocking the boat ramp. The ability to distinguish small localized 24

differences establishes the sensitivity of the methods used herein and are discussed in more detail below (Sect. 3.4). 25

The discharge from overfilled reservoirs also has significant anomalous properties with respect to the algorithm, 26

e.g., Thermalito Afterbay, which receives the discharge from Lake Oroville, was sampled after the overfilling of the 27





parent water mass during the drought-breaking California wet 2016–2017 winter. The two sets of data were obtained

in different locations with the higher CDOM data obtained in a shallow marsh area.

The refilled lakes in Fig. 4 frequently exhibit larger anomalies with respect to the algorithm than hypersaline or

alkaline lakes, especially in terms of turbidity as determined by the $K_d$ ratio. This is because many of the refilled

lakes are overfilled, wherein the shore of the lake extends beyond the normal acreage of the lake (e.g., Washoe Lake

and Little Washoe Lake). In overfilled lakes, land that is not normally flooded is added as new lake bottom, and the

new acreage is a source of atypical constituents, either in composition or concentration. The refilling of a normally

dry endorheic basin, e.g., White Lake, wherein the flood waters and the reclaimed lake bottom provide the maximum

areal and volumetric source of dissolved and suspended constituents results in some of the most extreme results,

both in terms of turbidity and with respect to the algorithm.

## 3.2 River Mouth, Resuspension, and Ice Edge Proximity

The inflow of dissolved and suspended constituents to a parent water mass is explored further by considering

a variety of sources that can provide absorption or scattering anomalies. The new data are shown in Fig. 5 and

were obtained in river mouths, water bodies with known suspension or visible resuspension, plus samples obtained

closest to or farthest from the ice edge within an oceanic ice field. Water bodies with known suspension or visible

resuspension are primarily from tidal and riverine flows, which are shown in Fig. 5 as triangles. Almost all of the

resuspension data were obtained at peak tidal flow to ensure safe navigation in the necessarily shallow waters. The

Akkeshi Bay data were obtained the day after the passage of typhoon Vongfong, wherein the shallow bay waters

were a distinctly different color than normal. The Sacramento River data were obtained after heavy rains, wherein

the boat ramp to be used was closed due to flood waters.

The resuspension data in Fig. 5 also include Bear Lake (which straddles the Utah-Idaho border) plus the effects

of a large ship docking in the shallow RWC Channel with the aid of a tug boat. The latter involved two vessels

churning up bottom material that significantly changed the color of the water. The resuspension sampling occurred

shortly after the ship was secured and the tug boat departed. The Bear Lake scattering anomaly is created primarily

through ground water seepage, which is rich in calcium carbonate particles (Davis and Milligan 2011). The ground

water is nutrient poor and the small amount of riverine input to the lake is through a swamp and wetlands, wherein

plants consume the nutrients and sediments settle out. Consequently, the Bear Lake data in Fig. 5 represent a

significant clear-water scattering anomaly with respect to the algorithm.



The other clear-water data in Fig. 5 were obtained principally in Arctic oceanic ice fields. These data were collected in two groups, which are distinguished as being closer to the ice and farther from the ice. These data almost always are displaced above or below the algorithm, respectively, even in the more turbid waters of Kotzebue Sound. The majority of these data were obtained by launching a small boat from the larger ice breaker used to safely enter the ice field, so the data obtained closer to the ice are usually as close to the ice as possible while being beyond the shading of the water mass by the ice field. The classification of *closer to* and *farther from* is qualitative and in complicated ice fields misclassification of the latter is possible. The Fig. 5 data show only two *farther from* points that are likely not classified correctly, and this category is the most vulnerable to a qualitative error.

In regards to the resuspension anomalies in Fig. 5, which all cluster below the algorithm, the river mouth data are the opposite—the data cluster above—but the number of river mouth observations is much smaller. The reduced number of observations is due to the difficulty of operating a trailered small boat in a shallow river, and then safely navigating the vessel out into the river mouth through a frequently narrow channel, wherein the higher sea state of the coastal ocean can be significantly amplified and boat traffic can make station work hazardous. Within the plume of a river mouth, two usually rather different water masses meet and mix over short time scales. Under those conditions, short-term anomalous water masses, with respect to the algorithm, can emerge and that is what is shown in the Fig. 5 data. Some of the data, however, are only mildly anomalous, which is also possible if the mixing has occurred over longer time periods, which might be tidal dependent.

### 3.3 Atypical Blooms, Invasive Species, and Harbors

The presence of an atypical bloom, particularly a harmful algal bloom (HAB), in a water mass, was anticipated to create an anomaly with respect to the original algorithm, because one or more significant stressors are frequently involved, e.g., an overabundance of nutrients (nitrates, ammonia, urea, or phosphate), which can be anthropogenic in origin (Heisler et al. 2008). In this context, an atypical bloom includes the concentration of an otherwise typical bloom to artificially extreme levels due to local weather, e.g., strong winds and waves increasing the amounts of algae on one side of a lake. Harbors and invasive species were also expected to frequently result in anomalies.

The new data obtained in harbors and water bodies experiencing an invasive species, or atypical bloom, including a HAB, are shown in Fig. 6. Some of these data could have had two classifications. For example, the Tahoe Keys and Tahoe Yacht Club were both infested with an invasive aquatic plant. Limited presence in one harbor and mechanical removal in the other implied a harbor classification was nonetheless appropriate. The Willamette River data were





from an area with an invasive aquatic species (Bierly et al. 2015) and the anomaly with respect to the algorithm 1
is opposite from the Lake Tahoe harbors. The latter suggests the classification for the Lake Tahoe harbors, which 2
cluster with the other harbors that were sampled, is likely appropriate. 3

Almost all harbors exhibit elevated $a_{\mathrm{CDOM}}(440)$ values with respect to the adjacent parent water mass, e.g., 4
Chula Vista, Treasure Isle, San Leandro, America's Cup, plus Tahoe Keys and Tahoe Yacht Club. The range of 5
harbor anomalies with respect to the algorithm has few extreme values, which is expected because harbors exchange 6
water with the parent water mass. San Leandro and El Granada have the largest anomalies, but San Leandro is 7
in a heavily urbanized area immediately south of Oakland International Airport in the San Francisco Bay area, so 8
significant anthropogenic sources are anticipated. 9

Like some coastal harbors, El Granada vessels are moored in an inner shallow harbor protected by an outer 10
deeper area, both with perimeter breakwaters. The two harbor areas cannot exchange water completely, i.e., a 11
portion of the water volume is trapped during each tidal cycle, and are more turbid than the parent water mass, Half 12
Moon Bay. The inner harbor is a likely and persistent anthropogenic source with a longer residence time, so it is 13
anticipated to have an $a_{\mathrm{CDOM}}(440)$ value that exceeds the neighboring bay. The outer harbor interacts tidally with 14
the inner harbor and the bay through narrow openings. The increased residence time and reduced exchange rates 15
are a possible mechanism to increase $a_{\mathrm{CDOM}}(440)$. This mechanism is also present in drought-stricken lakes. Other 16
harbors wherein a protected moorage has elevated $a_{\mathrm{CDOM}}(440)$ include Las Vegas (Lake Mead) and Crescent City. 17

The HAB data in Fig. 6 were frequently obtained opportunistically and, thus, were not necessarily from the peak 18
of the phenomenon. Also, a bloom is heterogeneous and navigation within the bloom is mostly based on visual 19
observations, so the anomaly with respect to the algorithm is not always extreme. The Monterey Bay HAB data 20
are the most extensive, because there was the opportunity for scheduling some of the data collection during a time 21
period when a HAB was likely to occur. In all cases, a HAB observation has a larger $K_d$ ratio than the algorithm 22
predicts, and this is principally caused by an increase in the $K_d(320)$ value, i.e., increased attenuation in the UV. 23

An atypical bloom is primarily a combination of local reports of atypical conditions, and a heterogeneous eutrophic 24
water mass, i.e., the chlorophyll concentration exceeds $1\,\mathrm{mg\,m^{-3}}$, with some water bodies having concentrations 25
greater than $10\,\mathrm{mg\,m^{-3}}$. Consequently, the lack of sophistication and specificity related to explaining the atypical 26
part of this subcategory does not exclude a simpler explanation. For example, local wind conditions could elevate 27
the values associated with a typical bloom into atypical concentrations. This phenomenon was observed in more 28
than one lake, e.g., Pyramid Lake and Upper Klamath Lake, and might have been missed for some of the atypical 29





data in Fig. 6. The majority of the atypical data are in rather close agreement with the algorithm, and as is shown 1
below (Sect. 4.), these data can be included in validating the algorithm with negligible consequences. 2

## 3.4 Wetlands, Pollution, and Water Mass Modifiers

The new data obtained in a wetland or polluted water mass are presented in Fig. 7. The wetlands are almost all 3
marsh grass except two, and these are labeled as to their types. The two unlabeled at the top of the plot are from 4
Cutoff Slough in California and are from a marsh grass. All wetlands exhibit the same anomaly, that is, they are all 5
displaced above the algorithm, although four are in rather close agreement with the algorithm. The polluted water 6
masses are associated with agricultural (Upper Klamath Lake and Upper Elkhorn Slough) or mining (Clear Lake) 7
runoff, with the latter being the most severe. For both Upper Klamath Lake and Clear Lake, blue-green algae were 8
plainly visible with extreme maximum chlorophyll concentrations of $1.117\,\mathrm{g\,m^{-3}}$ and $1.420\,\mathrm{g\,m^{-3}}$. The chlorophyll 9
concentrations in Upper Elkhorn Slough are less, but are still extreme with a maximum value exceeding $100\,\mathrm{mg\,m^{-3}}$. 10

Figure 7 also includes examples of a small inflow from a creek or another anomalous source modifying a parent 11
water mass, i.e., the much larger surrounding water. These data provide a measure of the sensitivity of the data 12
acquisition, processing, and analysis techniques used for this study. Although other sensitivity examples are docu- 13
mented above, e.g., the distinction between sampling closer to, or farther from, the ice edge in an oceanic ice field 14
(Fig. 5), the examples in Fig. 7 are subtler. Most of the modfiers are from creek inflows to the parent water mass, 15
but two are not. For the latter, the anomalous data are from a fish kill in the Salton Sea and a large floating and 16
partially submerged debris field in Shasta Lake after it refilled during the drought-breaking California wet 2016–2017 17
winter. In all cases, the anticipated anomalies appear different than the parent water mass. The water properties of 18
the creek inflow are not known, because access to the source from a small boat was problematic. 19

The generalized properties of the inflowing creek waters, determined visually, are as follows: a) the Lake Tahoe 20
inflow was turbid, milky meltwater from snow and ice melting on shorelands; b) the Shasta Lake inflow was from 21
rocky, tree-covered terrain and was significantly clearer than the lake water (the water pooled into a small pond before 22
flowing into the lake and was easily observed); c) the Donner Lake inflow was from a rocky, tree-lined canyon; d) the 23
Mono Lake inflow was across a mostly barren, rock-strewn shore with loose soil and was notably brown compared 24
to the green lake; and e) the Pinto Lake inflow was from a densely vegetated buffer zone adjacent to farmland. The 25
displacement of the anomalous waters with respect to the parent water mass are in keeping with these observations, 26
i.e., the waters subjected to turbid or clear inflows had larger or smaller $K_d(320)/K_d(780)$ ratios, respectively. 27



### 3.5 Alternative Spectral End Members

The end-member wavelengths used in alternative $K_d(\lambda_1)/K_d(\lambda_2)$ ratios, hereafter $\Lambda_{\lambda_2}^{\lambda_1}$, follow the combinations first used by Hooker et al. (2013), i.e., the UV-NIR $\Lambda_{710}^{340}$ pair, as well as the VIS $\Lambda_{670}^{412}$ pair. Shortly after the start of this study, C-OPS system 021 was upgraded (Table 1), so the $\Lambda_{875}^{313}$ pair is also available and provides the widest spectral span (562 nm) between end members. A plot of the end-member combinations is presented in Fig. 8, which also includes the linear fits and the root mean square error (RMSE) of the data with respect to the fits. The data in Fig. 8 are only those observations provided in Fig. 3, i.e., all 15 subcategories established in Sect. 2.5 (Figs. 4–7) are excluded. The consequences of using an increasing number of all the observations are presented in Sect. 4.

The fits in Fig. 8 show the end-member pair with the best accuracy is $\Lambda_{780}^{320}$, although the $\Lambda_{875}^{313}$ and $\Lambda_{710}^{340}$ fits are also to within the calibration uncertainty of the radiometers plus inevitable environmental variance for a net uncertainty to within 5%. The slope of the $\Lambda_{780}^{320}$ fit using the new validation quality data is indistinguishable from the original Hooker et al. (2013) algorithm ($y = 0.2556x - 0.0030$) and the slope agrees to within 1.1%. In general, as the end-member wavelengths are brought spectrally closer together, the variance in the Fig. 8 data increases and reaches a maximum for the $\Lambda_{670}^{412}$ pair, which degrades accuracy (the RMSE increases with decreasing spectral separation of the end members). In the case of $\Lambda_{875}^{313}$, the RMSE is a little larger than for $\Lambda_{780}^{320}$, but a little less than for $\Lambda_{710}^{340}$. The fewer number of $\Lambda_{875}^{313}$ data pairs creates gaps in the distribution of the data, as seen in Fig. 8, which partially explains why these data do not yield the lowest RMSE.

The $\Lambda_{875}^{313}$ Fig. 8 data show the variance also increases after the transition from more turbid to clear waters, i.e., $a_{\mathrm{CDOM}}(440) = 0.02\,\mathrm{m}^{-1}$, and the variance continues to increase with increasing water clarity. The larger variance as a function of water clarity is caused by the increasing importance of wave-focusing effects coupled with the increasing attenuation at 875 nm. Both of these problems are tractable for the $\Lambda_{780}^{320}$ end members, but contribute to the difficulty of deriving the data products and ultimately producing a stable $K_d$ ratio for the $\Lambda_{875}^{313}$ end members.

The increased $\Lambda_{710}^{340}$ and $\Lambda_{670}^{412}$ variances are not restricted to the problem described for $\Lambda_{875}^{313}$ end members. As end members are brought spectrally closer together, the range of expression available to distinguish two similar but optically different water masses decreases. Consequently, choosing the extrapolation interval is more sensitive to small changes in the parameters that ultimately determine the fit for the extrapolation interval. For legacy end members, clear waters have a lesser range of expression and turbid waters have the greatest, so this problem decreases as turbidity increases, which is seen in the $\Lambda_{710}^{340}$ and $\Lambda_{670}^{412}$ Fig. 8 data as a decrease in variance as turbidity increases.



### 3.6 Legacy Data Archive

The NASA bio-Optical Marine Algorithm Dataset (NOMAD) v2.a (Werdell and Bailey 2005) is used to determine  1

if the algorithmic approach used by Hooker et al. (2013) and adopted here can be independently confirmed. The  2

NOMAD database is a small, quality controlled subset of a larger data repository established early in the Sea-viewing  3

Wide Field-of-view Sensor (SeaWiFS) satellite mission (Hooker and Esaias 1993) called the SeaWiFS Bio-optical  4

Archive and Storage System (SeaBASS) and is described by Hooker et al. (1994).  5

The NOMAD database does not include applicable $a_{\mathrm{CDOM}}(440)$ measurements with contemporaneous UV and  6

NIR spectral end members as used in this study. Consequently, the Hooker et al. (2013) algorithmic approach, which  7

is based on UV and NIR end members, cannot be evaluated with NOMAD. The NOMAD database, however, does  8

include $K_d(\lambda)$ with legacy VIS wavelengths plus matching dissolved (Gelbstoff) spectral absorption coefficient at  9

443 nm, $a_g(443)$, which is functionally equivalent to $a_{\mathrm{CDOM}}(443)$ following Röttgers and Doerffer (2007).  10

The consequences of the 3 nm shift in $a_g(443)$ with respect to $a_{\mathrm{CDOM}}(440)$ are considered negligible for a general-  11

ized inquiry involving legacy optical data, because the fixed wavelengths in the radiometers have 10 nm bandwidths  12

and there are multiple sources of uncertainties in the derived optical data products of equal or greater importance  13

(Hooker et al. 2013), e.g., pressure tares, aperture offsets, dark current corrections, wave-focusing effects, etc.  14

From the full set of 4,459 NOMAD stations, 227 include $\Lambda_{670}^{412}$ end members and $a_g(443)$ observations, hereafter  15

$a_{\mathrm{CDOM}}(440)$, but 2 are duplicates. Application of $\Lambda_{670}^{412}$ data to the corresponding algorithm in Fig. 8 results in 13  16

observations with negative (predicted) $a_{\mathrm{CDOM}}(440)$ values, which are removed to leave 212 unique stations. This  17

process demonstrates how end-member algorithms can be used to quality assure optical data in archives (Sect. 3.7).  18

Another quality assurance procedure that could be used if the data were not restricted to legacy end members is to  19

test for compliance with all the end-member algorithms presented in Fig. 8. If an end-member pair does not produce  20

reasonable agreement with the applicable $a_{\mathrm{CDOM}}(440)$ algorithm (e.g., to within 5%), the spectrum could be flagged  21

as suspect. Such a test could also be used during optical data processing to determine the extrapolation interval  22

objectively by converging the agreement of all applicable end-member algorithms.  23

The 212 retained NOMAD stations are substantially—but not exclusively—located within the Chesapeake Bay  24

and its outflow into the surrounding southern Mid-Atlantic Bight, for which there are 189 data points, i.e., 89% of  25

the data are from a restricted geographic area. For the remainder, 13 are off the mouth of Delaware Bay, 9 are from  26

Massachusetts Bay, and 1 is in the open ocean northeast of South America. The southern Mid-Atlantic Bight and  27

parts of Massachusetts Bay were part of the data used by Hooker et al. (2013) to validate the original algorithm, so  28



data from these areas are anticipated to be compliant with the end-member algorithm. 1

The average depth of Chesapeake Bay is relatively shallow (approximately 6.4 m) with a signifiant portion (over 2
24%) less than 2 m deep. Given the extensive contribution of rivers, tributaries, and tides to bay dynamics, resus- 3
pension of material is anticipated to be a source of bias in optical properties with respect to end-member algorithms 4
for some bay stations (as shown in Fig. 5). 5

The retained NOMAD data are separated into two regimes: the north Chesapeake Bay (NCB) and all other 6
water masses, which consist of 106 stations for each. The dividing line for the NCB is the latitude of the Wicomico 7
River in the Maryland Eastern Shore (slightly north of the Potomac River mouth). The separation is arbitrary and 8
is used to compare the 106 NCB observations from NOMAD with 174 C-OPS $K_d$ ratios and $a_{\mathrm{CDOM}}(440)$ data pairs 9
obtained in the NCB (not shown in Fig. 2), albeit at different times and locations than the NOMAD data. During 10
data collection, the C-OPS sampling was with system 021 (Table 1) and included notations about *in situ* conditions 11
useful for establishing a resuspension subcategory, but the procedures predated and were not as rigorous as Sect. 2.5. 12

The C-OPS and NOMAD data plotted in Fig. 9 show general agreement (linearity) of the NOMAD data with 13
respect to the algorithm, which independently confirms the Hooker et al. (2013) algorithmic approach (and as 14
evaluated in more detail herein). Within the narrower turbidity range of the NOMAD and C-OPS NCB data 15
without likely resuspension, there is improved agreement. The C-OPS NCB resuspension data appear properly 16
categorized, because they are shifted correctly away from the algorithm (Fig. 5). There is evidence the C-OPS data 17
considered free of resuspension effects nonetheless include some resuspension (e.g., some solid circles in Fig. 9 extend 18
into the open circles as part of shallow-to-deep transects, thereby indicating the transect point in which resuspension 19
effects were assumed absent was likely premature). The NOMAD data exhibit a higher variance with respect to 20
the algorithm, which results in an increased RMSE of 37.8% (or 44.1% if the 13 omitted observations are included) 21
compared to the 6.2% value determined with C-OPS data (Fig. 8). The more extreme NOMAD values suggest a 22
subcategorization methodology that could be applied to archival data would improve agreement with the algorithm 23
(already demonstrated with the removal of 13 observations using the $\Lambda_{670}^{412}$ algorithm in Fig. 8). 24

If the NOMAD data are partitioned into turbid and clear subsets, using $a_{\mathrm{CDOM}}(440) > 0.2$ and $a_{\mathrm{CDOM}}(440) \leq 0.2$ 25
as thresholds, respectively, the fit equation for the turbid $\Lambda_{670}^{412}$ data is $y = 0.3437x - 0.2404$. The slope of this turbid 26
NOMAD fit is similar to the corresponding end-member fit presented in Fig. 8 for which $y = 0.3504x - 0.1033$, and 27
agree to within 1.9%. The fit for the clear NOMAD data, however, is $y = 0.0758x + 0.0648$, which is significantly 28
different at the 78.4% level. The 13 NOMAD stations that were not retained out of the original 225 NOMAD stations 29



were in clear waters, which is another indicator that the clearer NOMAD data are problematic. 1

With respect to the algorithm, the increased bias, variance, and 13 negative derived values obtained with NOMAD 2
data (which is a small, quality controlled subset of the larger NASA SeaBASS archive) in clearer waters suggests the 3
legacy data are degraded by sampling artifacts. Example degradations in legacy free-fall instruments include wave- 4
focusing effects (because of the slower sampling rates), coarser VSR (because of faster descent rates), and deeper 5
extrapolation intervals (because of near-surface data loss from large vertical tilts and large aperture depth offsets). 6

Although some of the problems with legacy data are almost completely absent from C-OPS data (e.g., because 7
there is no righting moment phenomenon when a C-OPS profiler begins sampling and C-PrOPS further stabilizes the 8
planar orientation of the radiometers), some aspects of these limitations are present in the Fig. 8 data, but they are 9
not significant, i.e., they result in a small increase in variance, which only slightly degrades algorithm performance. 10
Inclusion of the C-OPS NCB data without resuspension to the derivation of the $\Lambda_{670}^{412}$ algorithm (Fig. 8) results in 11
rather small changes to the fit coefficients. The slope is to within 4.3% and the intercept is to within 4.6%, both of 12
which are to within the net 5% uncertainty for calibration and environmental variance. 13

### 3.7 Objective versus Subjective Classification

The data set established herein has an extensive number of observations directly suitable for validation exercises 14
(Figs. 3 and 9) plus 15 subcategories (Sect. 2.5) of potentially (but not automatically) problematic water bodies 15
(Figs. 4–7), with the latter determined subjectively. The combination yields 16 categorizations of data spanning an 16
arguably *global* sampling of open ocean, coastal zone, and inland water masses in terms of a generalized perspective 17
of the dynamic range in water properties (Figs. 3–8)—albeit with an acknowledgment that an individual *in situ* 18
sampling activity as undertaken here is not *truly* global. The NOMAD search (Sect. 3.6), however, showed archival 19
data provided a significantly less global data set in terms of size, as well as, spectral and geospatial extent, i.e., there 20
were no data with the spectral expanse or dynamic range as used herein. 21

The NCB C-OPS data used to compare with the NOMAD data were collected prior to the observations obtained 22
for this study (Figs. 3–8) and included a single subjective subcategory based solely on resuspension. For both the 23
NCB and global C-OPS data sets, statistical and graphical evidence showed application of subcategories to filter the 24
data could improve the validation process as determined by fit coefficients and algorithm accuracy (e.g., slope and 25
RMSE, respectively). The results also showed the subcategories of data spanned a range of agreement with respect 26
to the original Hooker et al. (2013) algorithm (Figs. 4–7); some data agreed closely and others showed significant 27



disagreement, e.g., atypical blooms and HABs, respectively (Fig. 6). Subcategories with extensive sampling, e.g., 1

drought-stricken and refilled lakes (Fig. 4) exhibited a continuum of outcomes spanning good to poor agreement. 2

Archival data usually do not include a subcategory parameter for the station where observations were obtained, 3

e.g., NOMAD has no applicable keyword in file headers. Although some subcategories could be determined from 4

geolocation, temporal, and survey information (e.g., a harbor, wetland, alkaline or hypersaline lake, and likely a 5

drought-stricken lake), other influences are not usually established without an observer (e.g., atypical bloom or 6

resuspension caused by a vessel). Consequently, a subcategorization scheme based on the optical measurements 7

alone might be advantageous to the validation process, particularly for archival data. 8

The subcategory classification approach (Sect. 2.5) included careful inspection plus knowledge of one or more 9

significant constituents or stressors to assure data quality. The subcategory approach can be evaluated using the 10

$K_d(\lambda)$ spectra for the validation quality data in Figs. 3 and 8 plus the 15 already identified subcategories of potentially 11

problematic water masses presented in Figs. 4–7. The combination yields a total of 16 categorizations that can be 12

described based on their spectral shapes and magnitudes. A small number of observations are excluded from the 13

objective approach to ensure consistency in the determination of all $K_d(\lambda)$ values, e.g., the White Lake data had 14

estimated values in the UV domain, Bear Lake is a unique scattering anomaly created by calcium carbonate particles, 15

ship-induced resuspension is anthropogenic in origin, etc. With the additional restriction of wavelength commonality 16

spanning 320–780 nm (Table 1), a total of 1,171 spectra are used for the objective classification analysis. 17

Previous studies applied fuzzy c-means (FCM) classification to ocean color algorithm development (Moore et al. 18

2001) or uncertainty estimation (Moore et al. 2015), thereby demonstrating the usefulness of the FCM over a crips 19

or hard (e.g., k-means) classification. Application of the FCM classification to log-transformed $K_d(\lambda)$ data for the 20

16 subcategorizations successfully classifies all $K_d(\lambda)$ spectra into five classes based on the Calinski and Harabasz 21

(1974) index (Matsuoka et al. 2013). Spectral shapes, as well as their magnitudes, uniquely vary between the five 22

classes and span a continuum of water masses from oceanic and lacustrine case-1 to extreme case-2 inland waters. 23

The continuum of water mass composition is summarized by the centered spectra for the five FCM classes shown 24

in Fig. 10, where $N_i$ is the class number set by index $i$. The diversity achieved in sampling lacustrine water masses 25

is revealed in Fig. 10 by the range of $K_d(\lambda)$ spectra for the example drought-stricken and refilled lakes shown, which 26

are compared with Crater Lake and the $K_d$ values of pure seawater, denoted $K_w$ (Morel and Maritorena 2001). All 27

of the FCM classes are based on the water masses sampled herein, except $N_1$ includes the influence of $K_w$. The 28

refilled lakes are shown to emphasize the anomalies presented in Figs. 4–7 are not detectable by the $K_d(\lambda)$ spectrum 29





alone, but require an understanding of the applicable end-member ratio and the $a_{CDOM}(440)$ value.

The two lakes at the bottom and top of the dynamic range in Fig. 10, are Crater Lake and White Lake, respectively, with the latter being the largest anomaly in Figs. 4–7. Using $K_d^{-1}(\lambda)$ as a proxy for the vertical scale that must be properly sampled, i.e., a sufficient number of observations must be obtained within the vertical scale to derive data products, the vertical scale in Fig. 10 ranges from about 14 m (Crater Lake) to 3 mm (White Lake), with the latter only being possible with unprecedented VSR. The shorter wavelengths for White Lake are shown as "estimated" because these wavelengths had to be processed with individual extrapolation intervals to provide the $K_d$ estimates, which is not in keeping with all the other data products.

The average $K_d(313)$ value obtained for deep water (586 m) sampling in Crater Lake was $0.072\,\mathrm{m^{-1}}$ and the coefficient of variation from the six casts was 1.1%. According to Morel et al. (2007) Crater Lake is "extremely" clear water, and the CV shows exceptional reproducibility, i.e., 1% radiometry. Note that as shown in Fig. 3, Crater Lake was sampled twice. A station with higher $a_{CDOM}(440)$ values was conducted in shallower water above submerged moss that grows in large, dense mats. The constituent properties of the water were anticipated to be influenced by the moss matts, and the $a_{CDOM}(440)$ values are elevated with respect to the deeper station.

The proportionate composition of each FCM class in Fig. 10 as a function of the original subjective subcategories, wherein contributions less than 5% are reported but not considered significant, is presented in Table 2. Using Fig. 10 and Table 2, the corresponding principal class characteristics are as follows:

$N_1$  The $K_d(\lambda)$ spectra have minimum values in the blue domain (400–490 nm) and maximum values in the NIR. The spectral shape is consistent with typical case-1 waters. The proportional makeup is dominated by the validation quality subcategory (Figs. 3 and 8) at 83%. The only other principal contribution is from Farther from Ice data (Fig. 5) at 14%, which are almost exclusively from slightly modified case-1 waters.

$N_2$  The minimum values for $K_d(\lambda)$ spectra are shifted to longer wavelengths (490–565 nm) from modifications to case-1 conditions, principally from proximity to ice edge effects wherein the ice is a sufficient source to alter case-1 properties. The maximum values in the NIR spectral domain are similar to case-1 waters. The proportion of validation quality data decreases to 66% and is supplanted with similar contributions by the proximity to ice subcategories (Fig. 5), which represent slightly modified case-1 waters.

$N_3$  The $K_d(\lambda)$ minimum is near the middle of the green domain (555–565 nm) due to increasing optical complexity as case-2 constituents appear in larger proportions. The UV domain values are the same as, or slightly lower than, the NIR domain. The validation quality proportion decreases to 39% while case-2 subcategories increase



significantly, i.e., resuspension, drought-stricken and refilled lakes, harbors, and HABs.   1

$N_4$   The minimum $K_d(\lambda)$ values are shifted into the green-red domains (555–625 nm) with maximum values in   2

the UV exceeding the NIR. Optical complexity reaches a maximum, because all 16 categorizations contribute   3

at the 1% level or more. The proportion of the validation quality subcategory is the most abundant, but is   4

further decreased to 36%. The resuspension, harbors, and refilled lakes subcategories provide net increases in   5

case-2 waters with additional extreme contributions from alkaline and hypersaline lakes.   6

$N_5$   The $K_d(\lambda)$ minimum is shifted into the NIR domain (710 nm), and $K_d(\lambda)$ is distinguished by the highest   7

values compared to the other classes, with the peak in the UV. The resuspension subcategory is dominant at   8

38%, followed by drought-stricken lakes at 14%, and the validation quality subcategory is reduced to 13%. The   9

remaining principal contributors are case-2 waters, i.e., wetlands, refilled lakes, and polluted water masses.   10

The decrease in the percent composition of the validation quality data as a function of increasing class number ($N_1$–   11

$N_5$) is an indicator of the difficulty of validating an algorithm within increasingly complex waters. The recurring   12

contribution of a relatively small number of principal subjective subcategories to the gradient in optical complexity   13

starting with $N_2$ and then continuing for $N_3$–$N_5$ confirms the original subcategory approach has merit.   14

Table 2 also indicates the resolution or granularity of the 15 subcategories was perhaps more nuanced than   15

required. For example, alkaline and hypersaline lakes could be one subcategory, as could refilled and drought-stricken   16

lakes. Combining subcategories does not ensure an eventual convergence with the objective FCM classification,   17

because the latter partitions the processes present in the subcategories into varying degrees of contribution for each   18

identified class. This partitioning involves both direct and indirect evidence, which is perhaps best realized with the   19

original granularity of the 15 subcategories as revealed by considering resuspension processes.   20

The direct evidence of resuspension is provided by the resuspension subcategory (created for this phenomenon).   21

Indirect resuspension is present or likely present in multiple subcategories, however. For example, melt water releases   22

particles at the ice edge, bottom deposits in harbors are stirred up by boat propellers, wind and rain redeposit   23

exposed bottom material into drought-stricken lakes, refilled lakes contain suspended material from riverine inflow,   24

river mouths contain suspended material as part of the riverine outflow, tidal and wave action suspend material in   25

shallow wetlands, etc. The Table 2 data show the percent contribution of direct and indirect suspension increases with   26

increasing class number $N_1$ (17%) to $N_3$ (46%) to $N_5$ (77%). Consequently, a significant part of the cascade towards   27

complexity is correlated with water masses that are not evolving conservatively, e.g., as a result of resuspension.   28

This implies water masses not evolving conservatively can likely be identified using $K_d(\lambda)$ values determined from   29





in-water optical data or (ultimately) airborne observations without a need for laboratory analyses. 1

A small number of principal subcategories ($N_p$) significantly determine the $K_d(\lambda)$ classification spectra, although 2

the paucity of observations for some subcategories, in part opportunistic or planned depending on the subcategory, is 3

a largely unknown mitigating factor. Nonetheless, if refilled and drought-stricken lakes plus hypersaline and alkaline 4

lakes, are respectively considered one subcategory rather than two, which appears reasonable based on Fig. 4, five 5

or less principal subcategories determine 97%, 92%, 85%, 80%, and 96% composition for $N_1$–$N_5$, respectively. 6

Without the original subcategory scheme, the importance of direct or indirect resuspension would not have 7

emerged with the clarity provided in Table 2. The composition of the FCM classes confirm the applicability of 8

subcategories while revealing which ones are important to each class and which ones are marginally or not important. 9

For the latter, the parent water mass modifier is not significant, as expected, because this small subcategory was 10

created to demonstrate the sensitivity of the *in situ* optical and laboratory methods. Other subcategories might 11

appear insignificant because of the difficulty of obtaining the sample (e.g., river mouth is underrepresented) or the 12

paucity of opportunities to collect a sample (e.g., local authorities are mobilized to prevent invasive species). 13

## 4. Discussion

The optical data acquired herein had a VSR in near-surface waters less than 1 mm. This allowed the derivation of 14

data products at all wavelengths spanning 313–875 nm for a dynamic range encompassing a global perspective of water 15

masses as described by approximately three decades of generalized water properties. The validation approach was 16

based on the concept that water masses evolving conservatively (i.e., free from stressors that might cause anomalies 17

to the natural range in the gradient of a constituent) are suitable for validating the original Hooker et al. (2013) 18

inversion algorithm for deriving $a_{\mathrm{CDOM}}(440)$ from $K_d(\lambda)$ spectral end members. 19

The identification of 15 subjective subcategories that were likely not evolving conservatively yielded 609 validation 20

quality data points spanning extremely clear to highly turbid waters sampled within the open ocean, coastal zone, 21

and inland waters. The new data adhered to the original $\Lambda_{780}^{320}$ algorithm (Fig. 3) with an RPD of 0.02% and an 22

APD of 3.86%. Alternative spectral end members (e.g., $\Lambda_{875}^{313}$, $\Lambda_{710}^{340}$, and $\Lambda_{670}^{412}$) had increasingly larger RMSE values, 23

but were to within the calibration uncertainty of the radiometers plus inevitable environmental variance (i.e., a net 24

uncertainty to within 5%) except for the narrowest spectral span of end members, which was 6.2% for $\Lambda_{670}^{412}$ (Fig. 8). 25

Although no data archive exists with the spectral and spatial coverage used herein, NOMAD $\Lambda_{670}^{412}$ data showed 26

general agreement (linearity) with respect to the original Hooker et al. (2013) algorithm, and independently confirmed 27





the algorithmic approach. There was also general agreement between the NOMAD NCB data and the corresponding C-OPS NCB data without likely resuspension, wherein the C-OPS NCB resuspension data appeared properly categorized, because they were shifted in the correct direction away from the algorithm (Fig. 5). The importance of near-surface VSR, which is quantitative evidence of the successful mitigation of a variety of sampling difficulties (e.g., large aperture offsets, righting moment instabilities, wave focusing effects, etc.).

The high VSR achieved with C-OPS resulted in an increased sensitivity for deriving data products in turbid water masses with vertical scales as small as 3 mm (Fig. 10). This sensitivity allowed small localized changes in a parent water mass to be distinguished, e.g., the difference between sampling closer to, or farther from, an ocean ice edge (Fig. 5), as well as distinguishing the consequences to a parent water mass from a creek flowing into a cove, a large floating and partially submerged debris field, or a fish kill (Fig. 7). The laboratory analyses were similarly sensitive.

The ability to distinguish small differences in water properties ensured the large-scale discrimination of 15 subjective subcategories, wherein each subcategory represented a water mass that was not evolving conservatively and not considered appropriate for algorithm validation. Although initially applied subjectively, the majority of the 15 subcategories can be determined objectively using survey information (e.g., a chart) plus temporal or spatial data. For example, wetlands, river mouths, harbors, alkaline and hypersaline lakes, and ice edge proximity are identifiable using spatial data, whereas identification of likely areas of tidal resuspension, polluted water bodies, plus drought-stricken and refilled lakes require spatial and temporal data that are likely accessible. Atypical blooms and HABs are problematic, but the application of temporal and spatial analysis can categorize a portion of these phenomena.

Plots of the $a_{\mathrm{CDOM}}(440)$ and $\Lambda_{780}^{320}$ relationships for the 15 subcategories of potentially problematic water bodies revealed some data that were significantly anomalous with respect to the original algorithm and others that were not (Figs. 4–7). The wide range in anomalies is largely the result of the substantial effort that was made to adopt a global perspective and sample the greatest diversity of water bodies possible, some of which were very difficult to access and sometimes required shoreline launches of the optical instrumentation that C-PrOPS made possible, e.g., severely drought-stricken and refilled lakes (high and dry, flooded, or debris-blocked boat ramps) plus hypersaline and soda lakes (Mono Lake, Salton Sea, Borax Lake, and Soda Lake had no boat ramps).

The accuracy of the algorithm as a function of including increasing proportions of the 15 subcategories of water masses not necessarily appropriate for validation exercises (Figs. 4–7), because the water masses were likely not evolving conservatively, is explored by expanding the 609 validation quality observations of $\Lambda_{780}^{320}$ end members (Figs. 3 and 8) to include the following subcategories (hereafter referred to as the second data set): inflows to a parent water



mass that are not hypersaline or drought-stricken lakes, closer to or farther from ice edge proximity, river mouth, resuspension (but not including Bear Lake and the ship-induced RWC Channel resuspension), atypical blooms, HABs, and wetlands. This second data set has 930 observations, the linear fit is $y = 0.2511x - 0.0046$, the RMSE is 5.7%, and the new slope is to within 1.7% of the original value presented by Hooker et al. (2013), i.e., $y = 0.2556x - 0.0030$.

The reason the slope of the second data set is not significantly different than the original fit coefficients is the data that were added provide anomalies above and below the distribution of the validation quality data set, as shown in Figs. 4 and 5. The C-OPS NCB data without resuspension cluster on or below the algorithmic relationship (Fig. 9). If these data are added to the second data set used to derive the $\Lambda_{780}^{320}$ algorithm (the 320 nm and 780 nm wavelengths were always part of system 021 as shown in Table 1), the resulting slope and intercept is $y = 0.2561x - 0.0076$, which is to within 2.0% of the slope determined for the second data set ($y = 0.2511x - 0.0046$).

If a third data set is created by adding drought-stricken and refilled lakes to the second data set, but not including White Lake (which had some estimated data products), this third data set has 1,044 observations. The linear fit is $y = 0.2249x + 0.0044$, the RMSE is 6.8%, and the new slope is reduced (as expected, because the added data are all below the algorithmic relationship and primarily turbid (Fig. 4), but still within 10.4% of the original value presented by Hooker et al. (2013). The exclusion of White Lake, as well as hypersaline, alkaline, and polluted water bodies, from the third data set is for a practical reason: they are extreme water masses and White Lake rarely exists.

There are other lakes presented herein that can be considered extreme, e.g., other endorheic lakes (e.g., Pyramid Lake, Eagle Lake, etc.), plus shallow lakes in high wind areas wherein bottom material is resuspended on a near-continuous basis (e.g., Washoe Lake and Little Washoe Lake). If all subcategories, except extreme lacustrine water bodies, are used to create a fourth data set it has 1,086 observations, i.e., almost 90% of the 1,230 maximum and 93% of the data used in Table 2 to create the five objective FCM classifications. The linear fit of this fourth more comprehensive data set is $y = 0.2379x - 0.0049$, the RMSE is 6.2%, and the new slope is not reduced as much as in the third data set (as expected) and is to within 6.9% of the original value presented by Hooker et al. (2013).

With the exception of the third data set that added primarily turbid data exclusively below the algorithmic relationship (drought-stricken and refilled lakes), all of the results from the expanded data sets are rather indistinguishable from the original $\Lambda_{780}^{320}$ fit provided by Hooker et al. (2013), wherein $y = 0.2556x - 0.0030$, or the separate validation quality data set presented in Fig. 8, for which $y = 0.2583x - 0.0053$. Ignoring the third data set, the $x$-intercept for the expanded data sets is approximately equal to what can be expected for pure water, i.e., 0.02. The close agreement of the various expanded data sets with adherence to the pure-water limit is another significant





confirmation of the algorithmic approach using spectral end members.    1

This study used three different laboratory methods to determine $a_{\mathrm{CDOM}}(440)$ from water samples, and seven    2

different optical instrument suites to determine $K_d(\lambda)$. Despite the agreement between the $\Lambda_{780}^{320}$ fits and their $x$-    3

intercepts, the possibility the results are the result of an unidentified stochastic process has not been addressed. The    4

latter is not likely for the optical data, because the radiometers were calibrated at one facility and deployed with    5

strict adherence to the Protocols using the same acquisition software. Furthermore, data products were derived using    6

the same processing software with one operator, and the variance in optical data products is shown in Figs. 3–8.    7

The capabilities of the above- and in-water radiometers for C-OPS systems 021 (with C-PrOPS) and 039 (without    8

C-PrOPS) were intercompared to next-generation *hybridspectral* instruments (Hooker et al. 2018c) containing a    9

hyperspectral detector system plus 18 fixed wavelengths (system 038 in Table 1). The comparisons showed excellent    10

agreement, 4.2–4.8%, was obtained to within the calibration uncertainty (2.3–2.7%) plus natural variability (i.e., a    11

net uncertainty to within 5%), as long as the stability threshold for a backplane without thrusters or the noise limit    12

of the hyperspectral sensor was not exceeded (Hooker et al. 2018b and 2018c, respectively).    13

Although field data demonstrate small changes in parent water mass modifications can be discriminated (Fig. 7),    14

the laboratory methods and instruments were not systematically intercalibrated to establish an overall uncertainty.    15

In addition, the laboratory methods included different temperature controls and storage procedures for the water    16

samples, as well as application of null-point corrections in different spectral ranges. A lack of systematic intercali-    17

bration is not a significant detraction, because it is a common difficulty when combining observations from databases    18

(e.g., NOMAD data), and methods exist to nonetheless determine the efficacy of the combined data.    19

Following the technique established by Matsuoka et al. (2017), a statistical sensitivity analysis is used to examine    20

the uncertainty of the combined $a_{\mathrm{CDOM}}(440)$ values from the three different laboratory methods used in Figs. 3–    21

8. Briefly, for each optical and water sampling, normality of distribution for $a_{\mathrm{CDOM}}(440)$ was created using the    22

measured value as the mean ($\mu$) and 7% of the measured value as the standard deviation ($\sigma$). Similarly, normality    23

of distribution for $\Lambda_{780}^{320}$ was created using the measured value as $\mu$ and 5% as $\sigma$. A lower $\sigma$ percentage was used for    24

the latter, because the original data set retained each optical cast whereas only one water sample was obtained, so    25

the variability of the optical data was already significantly represented (Figs. 4–8).    26

For each original data pair, $10^5$ variations were prepared for both $\Lambda_{780}^{320}$ and $a_{\mathrm{CDOM}}(440)$. Of these data, $10^3$ were    27

randomly selected and the mean value was calculated for $\Lambda_{780}^{320}$ and $a_{\mathrm{CDOM}}(440)$ for each data pair. This exercise was    28

repeated $10^3$ times (bootstrap) and an overall $\mu$, and $\sigma$ were obtained. For the different combinations of applying    29





$\mu \pm \sigma$ to bound the dispersion of the original data, the resulting linear fits showed the new algorithm slopes changed

by 0.3–1.1% with respect to the $\Lambda_{780}^{320}$ validation quality data set presented in Fig. 8, i.e., $y = 0.2583x - 0.0053$. For

all combinations of $\mu \pm \sigma$, the maximum RMSE was 1.1%, which is similar to the 1.2% value presented in Fig. 8

for the $\Lambda_{780}^{320}$ validation quality data set. Consequently, the use of three different laboratory methods to determine

$a_{\mathrm{CDOM}}(440)$, which were not intercalibrated, does not significantly influence the results presented here.

Consequently, the state-of-the-art (approximately 1%) accuracy achieved with the $\Lambda_{780}^{320}$ end members (Fig. 8) is

due to the strict adherence to sampling and processing protocols coupled with unprecedented vertical resolution in

the optical data. This combination also resulted in all end-member combinations—including the legacy (visible) $\Lambda_{670}^{412}$

pair—having a superior accuracy compared to many common global inversion algorithms. For example, algorithms

for deriving $a_{\mathrm{CDOM}}(\lambda)$ based on $L_W(\lambda)$ and its normalized forms have an RMSE exceeding 10% or more (Mannino

et al. 2008 and 2014), and the ocean color (OC) variants of chlorophyll $a$ algorithms which use normalized $L_W(\lambda)$

band ratios (O'Reilly et al. 1998 and 2000) typically exceed 20% or more.

The robustness of the end member approach, is further confirmed by how the expanded (second, third, and

fourth) data sets, which had respectively increased amounts of water masses that were not evolving conservatively,

nonetheless yielded fits with RMSE values outperforming the aforementioned $L_W(\lambda)$ algorithms. In regards to which

of the variants best represents a global perspective, the $\Lambda_{\lambda_2}^{\lambda_1}$ pairs in Fig. 8 are considered the most appropriate.

If an algorithm is to be applied to a water mass that is not evolving conservatively, an individualized relationship

between $a_{\mathrm{CDOM}}(\lambda)$ and $\Lambda_{\lambda_2}^{\lambda_1}$ should be established, especially if the water mass is extreme, e.g., drought-stricken,

hypersaline, alkaline, etc. This will maintain the accuracy of the global relationship while providing a straightforward

mechanism for improving the study of nonconservative water bodies. As shown in Fig. 9, identification of water masses

that are not evolving conservatively can likely be determined using $K_d(\lambda)$ values determined from in-water optical

(e.g, C-OPS) data without a need for laboratory analyses.

With additional research to produce $K_d(\lambda)$ for all above-water wavelengths, e.g., expanding upon Cao et al.

(2014), the identification of water masses that are evolving conservatively or not can ultimately be made from

airborne or satellite observations. This determination allows for a sensitive indicator of water masses subjected to

atypical stresses that influence water quality, e.g., drought conditions. The onset of next-generation satellite data,

e.g., the Japanese Second-generation Global Imager (SGLI) mission (Honda et al. 2012), offers unique opportunities

for such monitoring because of the expanded spatial and spectral domain, with the latter allowing improved $\Lambda_{\lambda_2}^{\lambda_1}$

algorithm accuracy with respect to legacy missions, e.g., Moderate Resolution Imaging Spectroradiometer (MODIS).



## Acknowledgments

This work was principally supported by the National Aeronautics and Space Administration (NASA) as part of  1

planning for the Aerosol, Cloud, Ecosystems (ACE) satellite remote sensing mission. The next-generation perspective  2

benefitted from the anticipated calibration and validation activities of the SGLI, ACE, plus Plankton, Aerosol, Cloud,  3

ocean Ecosystem (PACE) missions. The high level of success achieved in the field work for those activities established  4

a foundation of understanding that was the direct consequence of contributions from individuals who contributed  5

unselfishly to the work involved (e.g., calibration, acquisition, processing, and sampling). The scientists included  6

(alphabetically) J. Brown, B. Hargreaves, T. Hirawake, T. Isada, R. Lind, J. Morrow, K. Negrey, and J. Nishioka;  7

their dedicated contributions are gratefully acknowledged.  8

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



**Table 1.** The nominal fixed wavelengths in nanometers, all with 10 nm bandwidths, for each optical profiling system as distinguished by serial number. The number of casts obtained is differentiated between backplanes without (C-OPS) and with (C-PrOPS) digital thrusters (note system 021 had both), wherein boldface numbers indicate the 15 wavelengths common to all profiling systems, e.g., 320 nm and 780 nm.

| Optical Profiling System | | | | | | | Number of Casts | |
| 010 | 014 | 021 | 021† | 034 | 038§ | 039 | C-OPS | C-PrOPS |
|---|---|---|---|---|---|---|---|---|
|  | 313 |  | 313 |  |  | 313 | 57 | 709 |
| 320 | 320 | 320 | 320 | 320 | 320 | 320 | **497** | **733** |
| 340 | 340 | 340 | 340 | 340 | 340 | 340 | **497** | **733** |
| 380 | 380 | 380 | 380 | 380 | 380 | 380 | **497** | **733** |
| 395 | 395 | 395 | 395 | 395 | 395 | 395 | **497** | **733** |
| 412 | 412 | 412 | 412 | 412 | 412 | 412 | **497** | **733** |
| 443 | 443 | 443 | 443 | 443 | 443 | 443 | **497** | **733** |
| 465 | 465 | 465 | 465 | 465 |  | 465 | 497 | 709 |
| 490 | 490 | 490 | 490 | 490 | 490 | 490 | **497** | **733** |
| 510 | 510 | 510 | 510 | 510 | 510 | 510 | **497** | **733** |
| 532 | 532 | 532 | 532 | 532 | 532 | 532 | **497** | **733** |
| 555 | 555 | 555 | 555 | 555 | 555 | 555 | **497** | **733** |
| 560 |  | 560 |  |  |  |  | 356 | 0 |
|  |  |  |  | 565 |  |  | 84 | 0 |
|  | 589 |  | 589 |  | 589 | 589 | 57 | 733 |
| 625 | 625 | 625 | 625 | 625 | 625 | 625 | **497** | **733** |
| 665 |  | 665 |  | 665 |  |  | 440 | 0 |
| 670 | 670 | 670 | 670 | 670 | 670 | 670 | **497** | **733** |
| 683 | 683 | 683 | 683 | 683 | 683 | 683 | **497** | **733** |
| 710 | 710 | 710 | 710 | 710 | 710 | 710 | **497** | **733** |
| 780 | 780 | 780 | 780 | 780 | 780 | 780 | **497** | **733** |
|  |  |  |  |  | 820 |  | 0 | 24 |
|  | 875 | 875 | 875 | 875 | 875 | 875 | 470 | 733 |

† Upgraded with C-PrOPS, a conductivity sensor, and new wavelengths.     § Hypbridspectral and equipped with C-PrOPS.





**Table 2.** The objective FCM classification of the data in Figs. 3–8 with a few omissions to ensure consistent data quality (e.g., White Lake, Bear Lake, ship-induced resuspension, etc.). The five classes $N_i$, where $i$ is the class number index, are shown with the number of $K_d(\lambda)$ spectra, $N_s$, within each class in parentheses, as well as the percent composition of the original 16 subjective subcategories equalling or exceeding a 1% contribution threshold for each class. The principal subcategories in each class, i.e., the most numerous (approximately 5% contribution or more), are shown in bold typeface. The number of principal subcategories ($N_p$) and the number of all subcategories ($N_a$) with a 1% composition or more are summarized at the bottom of the table in slanted typeface with the percent composition from all the principal subcategories shown in the last line.

| *Original Subcategory Name* | *Class Number and Composition* | | | | |
|---|---|---|---|---|---|
| | $N_1$ (244) | $N_2$ (263) | $N_3$ (305) | $N_4$ (265) | $N_5$ (94) |
| Validation Quality | **83%** | **66%** | **39%** | **36%** | **13%** |
| Farther from Ice | **14** | **10** | 1 | 2 | |
| Closer to Ice | 2 | **11** | 1 | 1 | |
| Resuspension | 1 | 1 | **10** | **16** | **38** |
| Refilled Lake | | 1 | **9** | **7** | **10** |
| Drought-Stricken Lake | | 4 | **8** | 3 | **14** |
| Harbor (or Marina) | | | **11** | **8** | 4 |
| Harmful Algal Bloom | | 1 | **8** | 2 | 1 |
| Wetland (or Marsh) | | | 3 | 3 | **11** |
| Polluted Water Mass | | | | 4 | **10** |
| Alkaline Lake | | | 2 | **5** | |
| Hypersaline Lake | | | 2 | **5** | |
| River Mouth | | | 3 | 3 | |
| Atypical Bloom | | 3 | 1 | 2 | |
| Parent Water Mass Modifier | | 2 | 3 | 2 | |
| Invasive Species | | | | 2 | |
| *$N_p$ ($N_a$) Subcategories* | *2(4)* | *3(9)* | *6(14)* | *6(16)* | *6(8)* |
| *$N_p$ Percent Composition* | **97%** | **87%** | **85%** | **77%** | **96%** |





Figure Captions

**Fig. 1.** The next-generation C-OPS backplane with C-PrOPS (roll is the long axis and pitch is the short axis into or   1

out of the page): a) irradiance cosine collector; b) radiometer bumper; c) array of 19 microradiometers; d) aggregator   2

and support electronics; e) rotating V-block for pitch adjustment; f) two-point harness attachment; g) *hydrobaric*   3

buoyancy chamber, which accommodates up to 3 compressible bladders; h) slotted flotation and i) bronze weights   4

for buoyancy and roll adjustment; j) water temperature probe and k) pressure transducer port on the radiance end   5

cap; l) conductivity sensor; m) electronics module; n) digital thruster (1 of 2, on each side); and o) thruster guard.   6

The aggregator and support electronics control the 19 microradiometers as a single device. The side bumpers and   7

thruster guards protect the radiometers and digital thrusters from unanticipated side impacts, respectively.   8

**Fig. 2.** The geographical distribution of the original data (solid diamonds) used in Hooker et al. (2013) versus the   9

new validation data (solid circles).   10

**Fig. 3.** The new validation quality data from the primary open ocean, coastal zone, and inland waters (blue, green,   11

and red circles, respectively) categories, which are used to evaluate the original Hooker et al. (2013) algorithm (gray   12

circles). A ±7.5% dispersion is approximately represented by the larger algorithm symbol size, i.e., from one edge   13

of a gray circle to the opposite edge represents a total of approximately 15.0% dispersion. The headwaters of the   14

San Francisco Bay Redwood Creek (RWC) Channel, which is surrounded by wetlands, is the most turbulent coastal   15

water mass and has the highest $a_{\mathrm{CDOM}}(440)$ value.   16

**Fig. 4.** The new data from lacustrine water bodies that were drought-stricken (magenta diamonds), refilled after   17

drought (blue triangles), alkaline (red squares), or hypersaline (green circles) in relation to the original algorithm   18

(gray circles). Each data point represents a single water sample with multiple optical casts for each, which results in   19

a series of results (typically 3–6) along the $x$-axis.   20

**Fig. 5.** The new data obtained in river mouths (red circles), water bodies with known suspension or visible resus-   21

pension (green triangles), plus samples obtained closer to (magenta squares) or farther from (blue diamonds) the   22

ice edge within an ice field. The San Francisco RWC Channel headland waters are depicted in Fig. 3 as are the   23

Columbia and Umqua River data that are upstream of the river mouth.   24

**Fig. 6.** The new data obtained in a harbor (or marina), plus water bodies experiencing an invasive species, HAB,   25

or atypical bloom.   26





**Fig. 7.** The new data obtained in a wetland or polluted water mass plus comparisons between a parent water mass          1
and a creek inflow or another source of anomalous water properties.          2

**Fig. 8.** Four end-member algorithms to derive $a_{\mathrm{CDOM}}(440)$ from in-water optical observations with the accuracy of          3
each estimated using RMSE statistics.          4

**Fig. 9.** The adherence of NOMAD archival data to the $K_d(412)/K_d(670)$ algorithm shown in Fig. 8 (gray solid          5
circles) for the north Chesapeake Bay (NCB) and other Mid-Atlantic Bight locations (red and orange solid diamonds,          6
respectively). The NOMAD NCB data are compared to C-OPS NCB data obtained at different times and locations          7
with the latter separated into two categories, wherein one is likely subjected to bottom resuspension (light blue open          8
circles) and the other is not (dark blue solid circles).          9

**Fig. 10.** The centered $K_d(\lambda)$ spectra of the five classes ($N_i$) determined from an objective FCM classification of the          10
data presented in Figs. 3–8 with a few omissions for data consistency (e.g., White Lake, Bear Lake, ship-induced          11
resuspension, etc.) and shown with respect to $K_w$. Example $K_d$ spectra from drought-stricken and refilled lakes plus          12
Crater Lake, obtained by averaging the results from multiple optical casts, are also shown to demonstrate the more          13
than three decades of dynamic range in turbidity that was sampled. The shorter wavelengths for White Lake (open          14
dark red squares) required individual wavelength processing to provide the estimated $K_d$ values, whereas all other          15
data were obtained with a single processing.          16



FIGURE 1

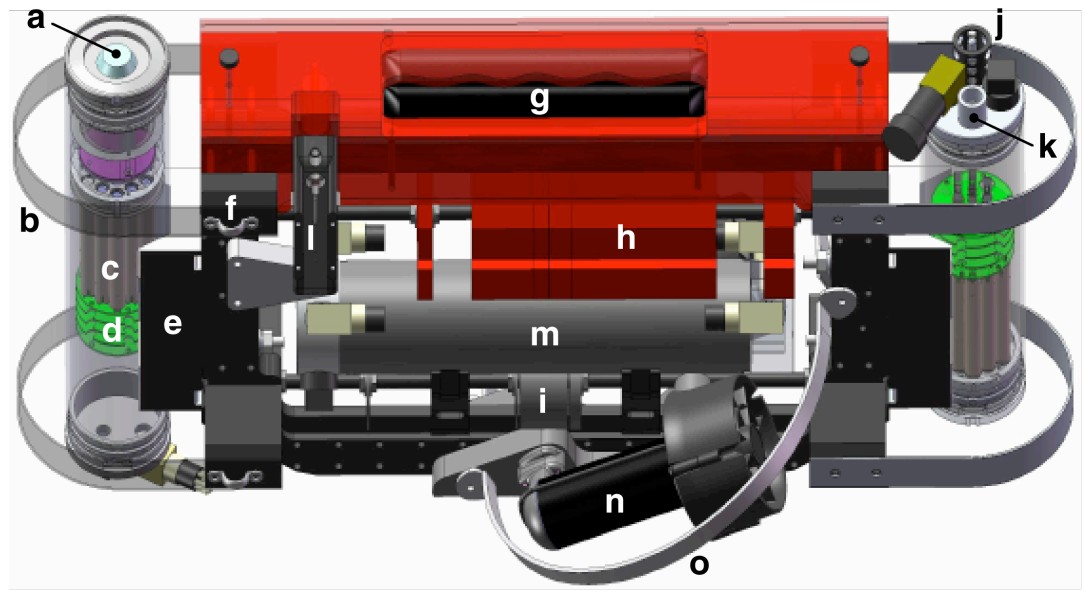





Figure 2



FIGURE 3

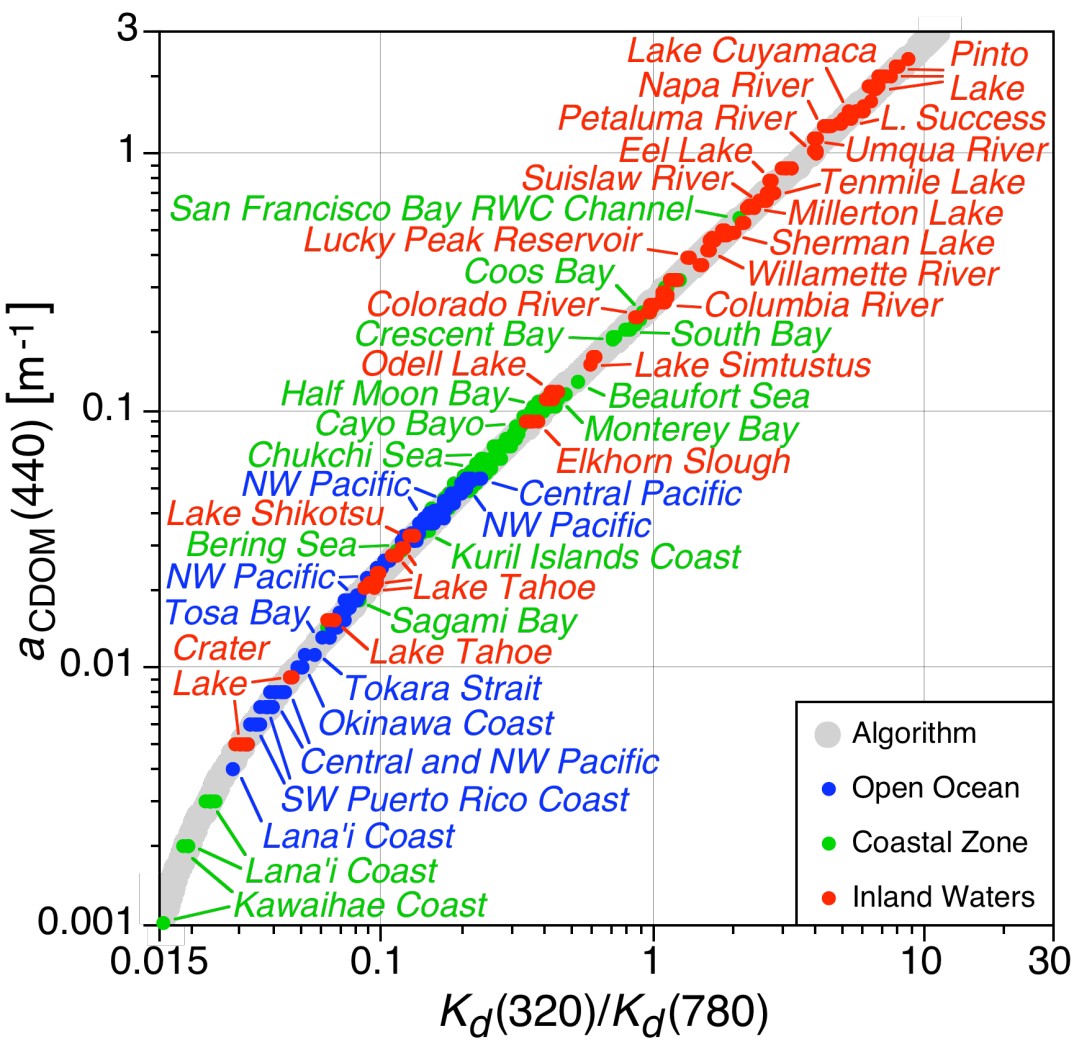



FIGURE 4

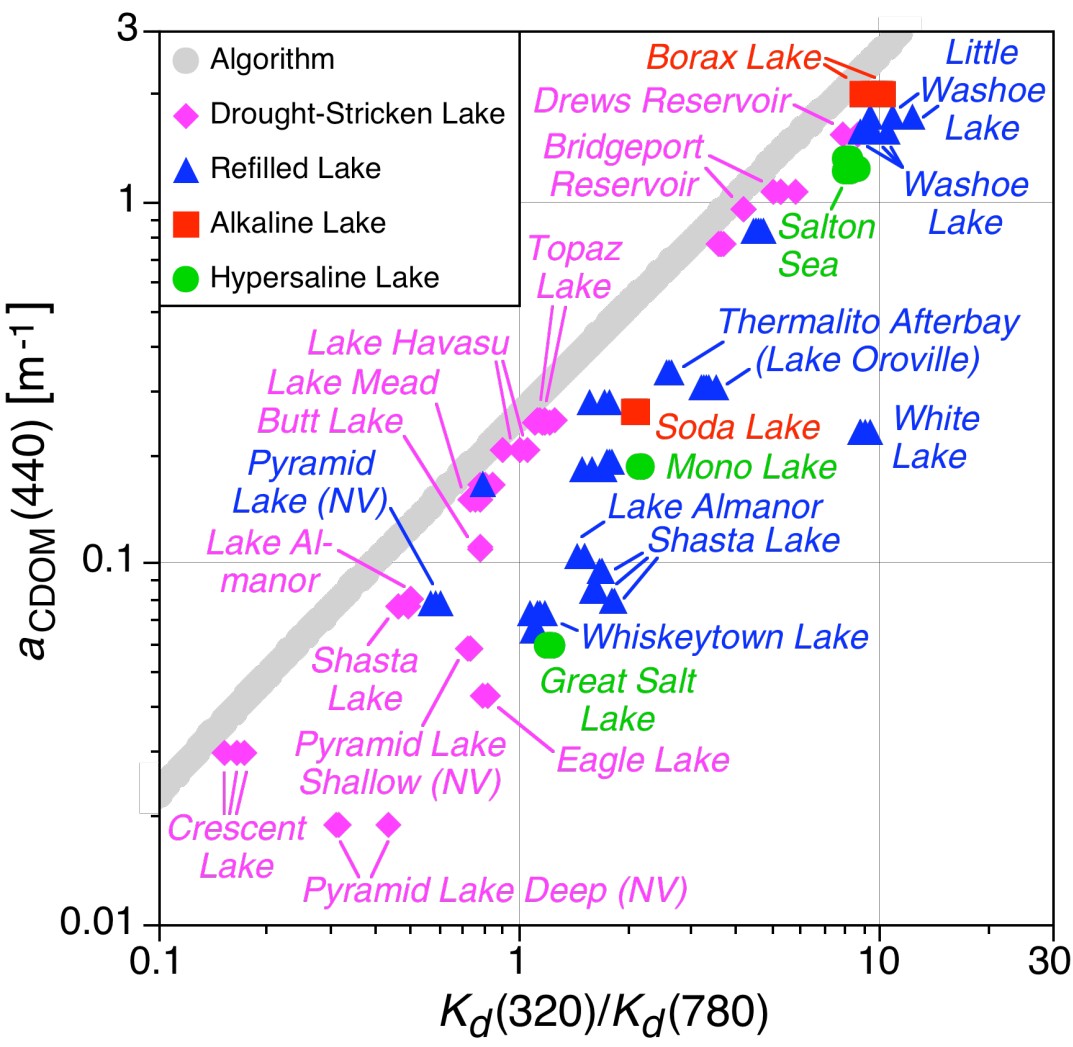



FIGURE 5

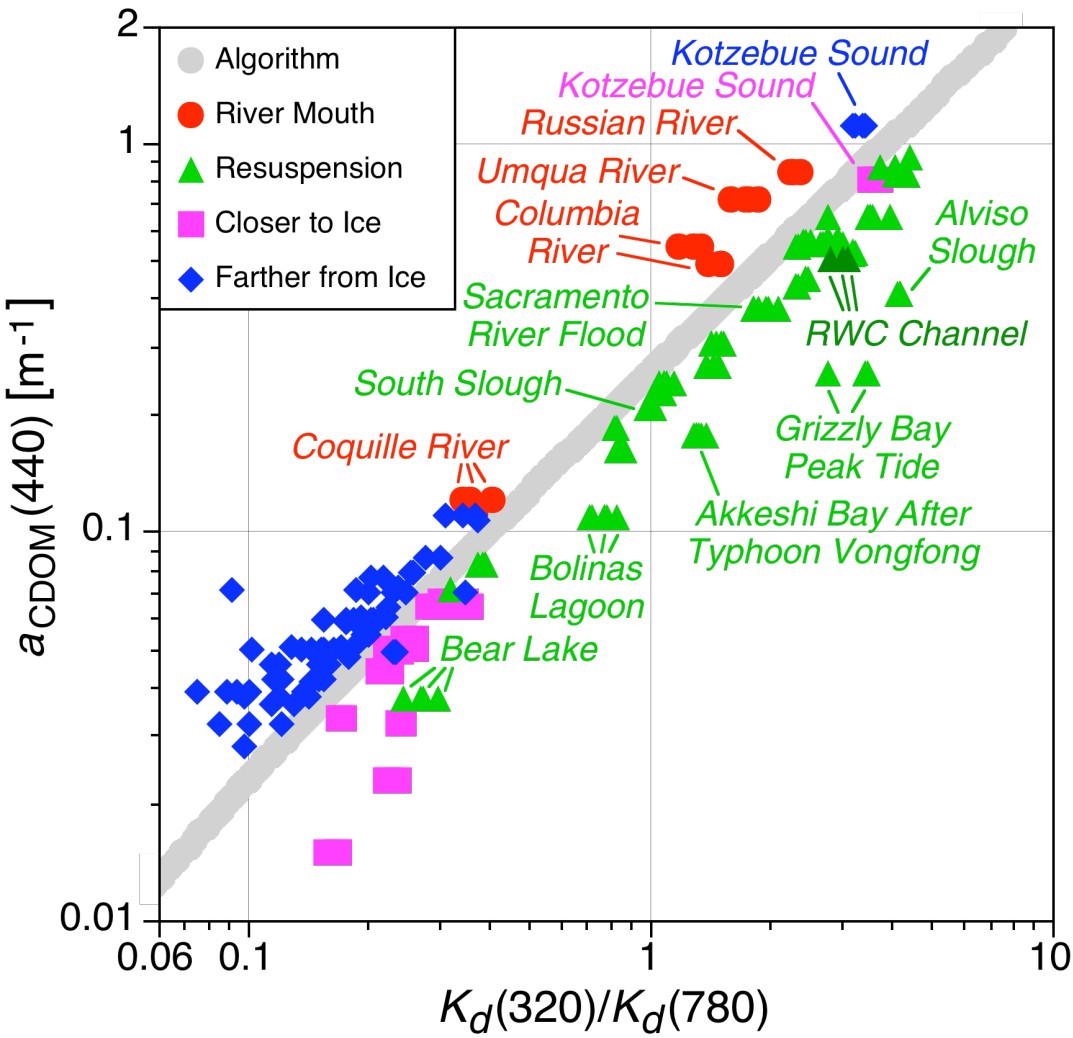




FIGURE 6

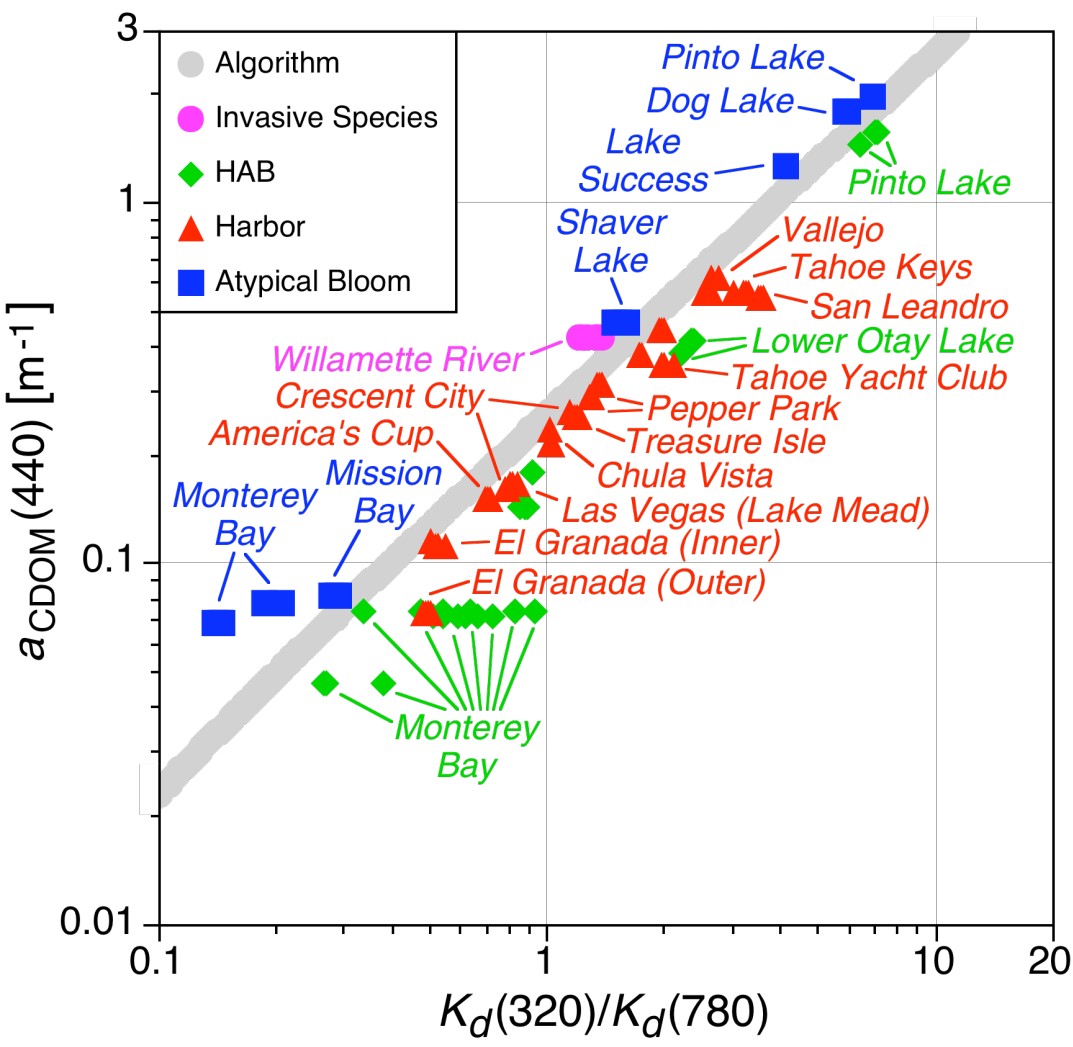



FIGURE 7

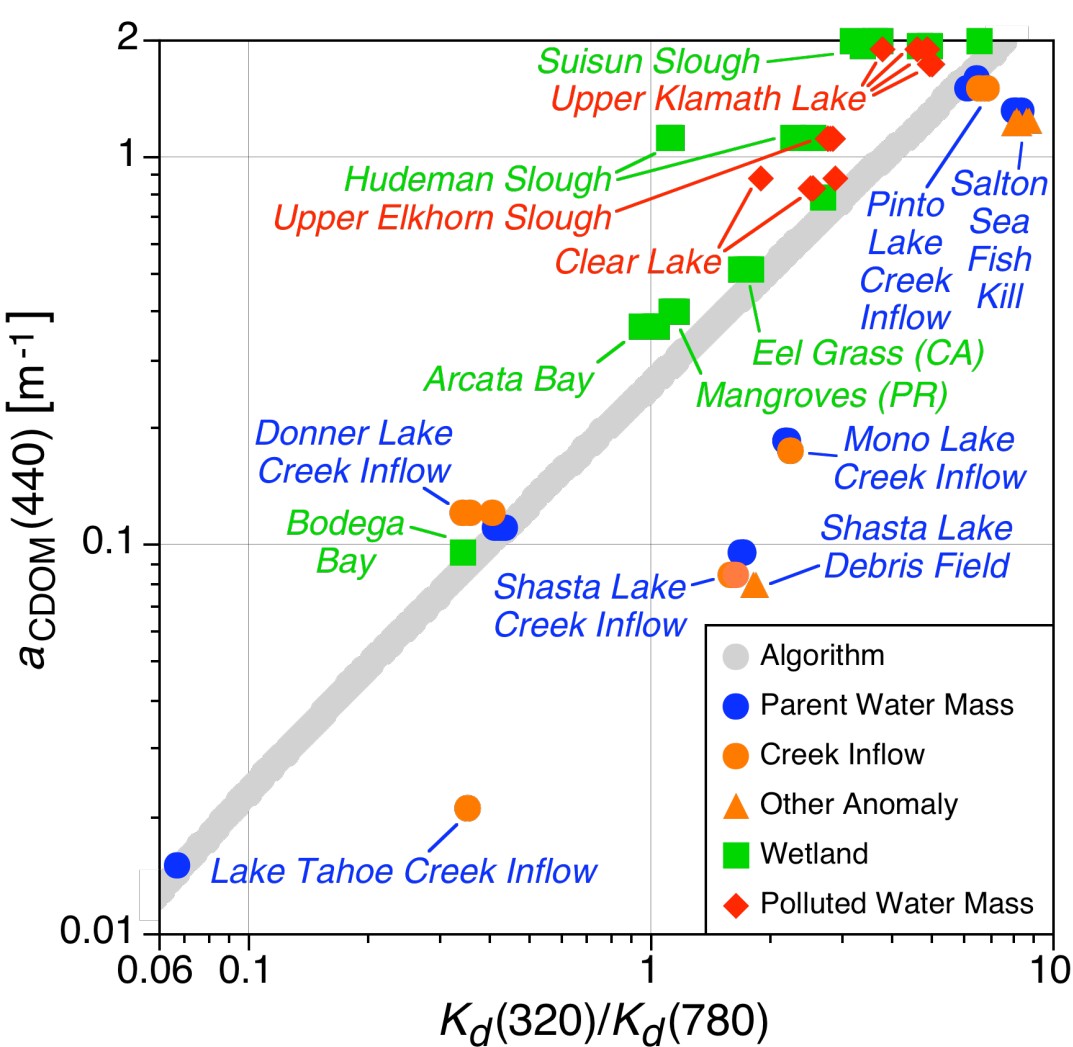


FIGURE 8

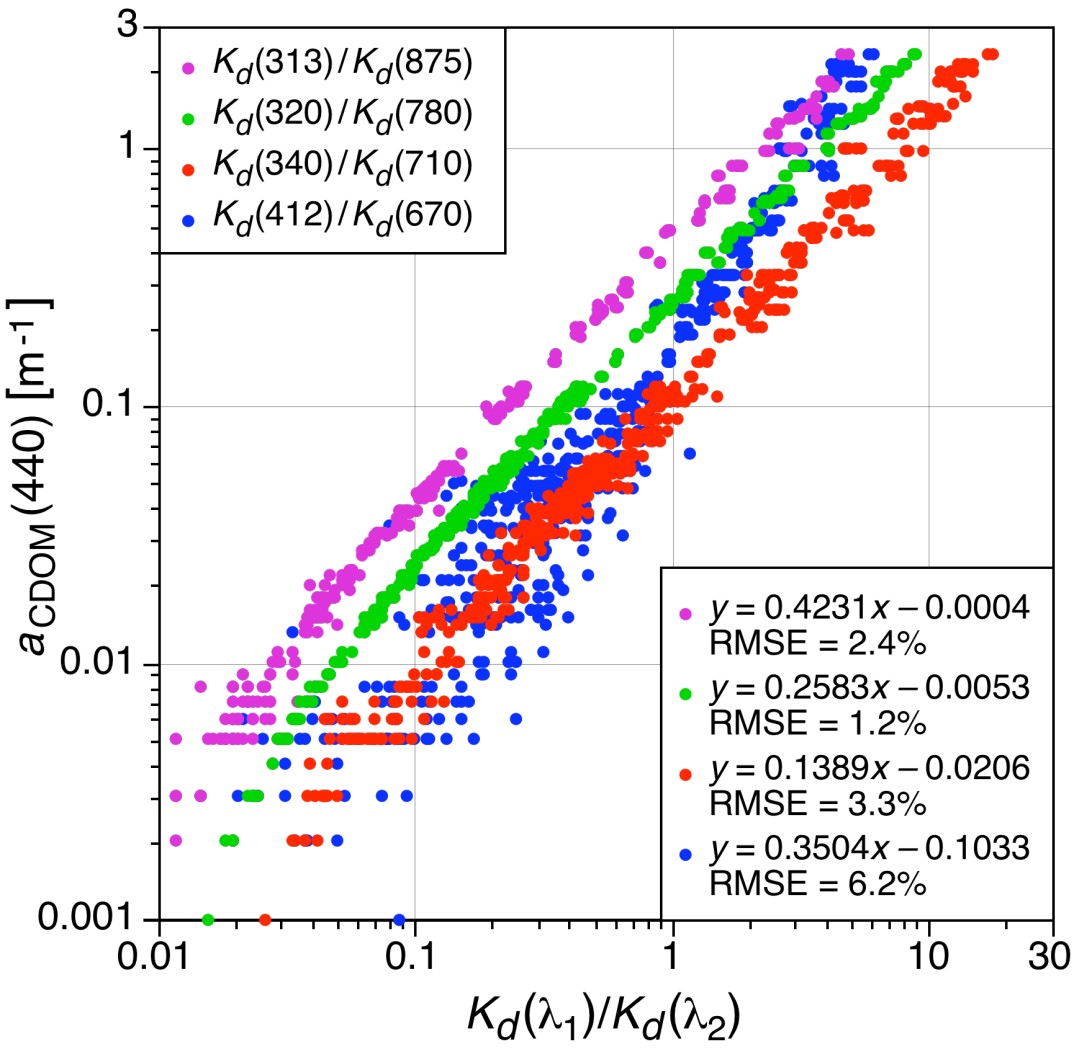



FIGURE 9

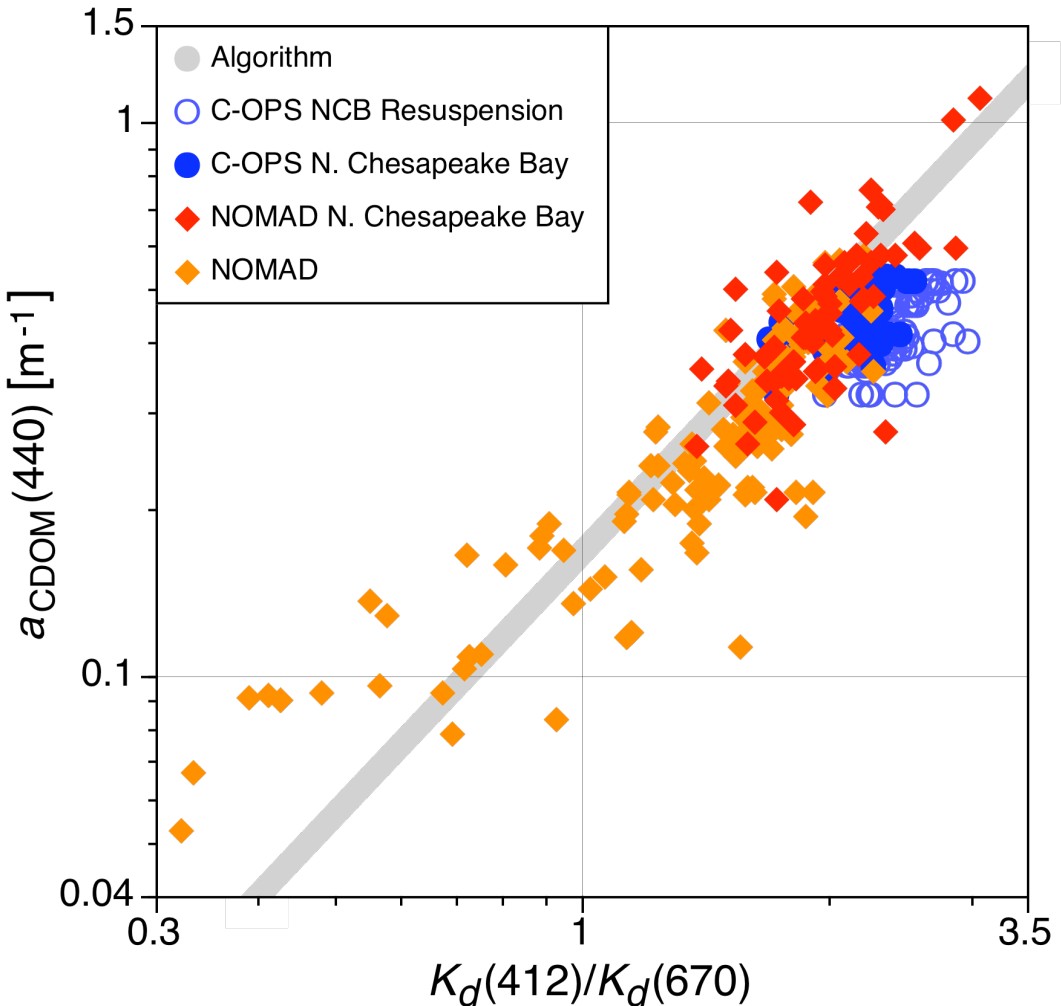



FIGURE 10

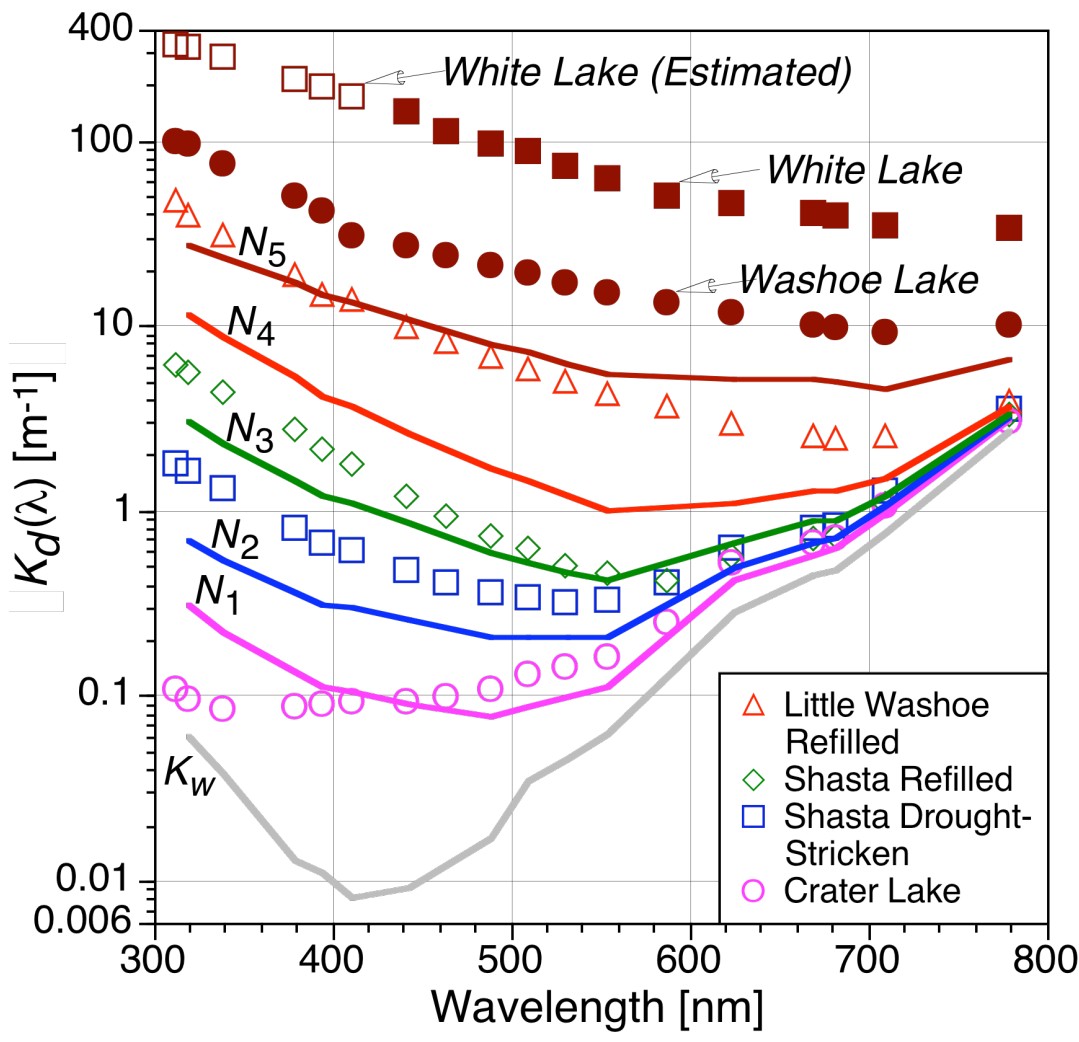