# Peer review of "A Global End-Member Approach to Derive $a_{\rm CDOM}(440)$ from Near-Surface Optical Measurements"

_Biogeosciences, 2019_

## Referee Comment (RC1) · Anonymous Referee #1 · 11 Aug 2019

Hooker et al. assess the performance of a Kd ratio-based empirical acdom(440) algorithm developed in Hooker et al. (2013) by using a large in-situ dataset collected across different water types ranging from turbid inland waters to clear oceanic waters. The use of spectral end member Kd ratios (Kd(320)/Kd(780)) is an interesting approach to estimate acdom(440). The authors extend the measurement suite (Beaufort Sea, Mackenzie River outflow and the Southern Mid-Atlantic Bight; Hooker et al. 2013) to a global perspective by sampling "extremely clear to turbid waters (e.g., North Pacific Ocean, the Arctic Ocean, Hawaii, Japan, Puerto Rico, and US east and west coasts) that span three decades of dynamic range in optical properties. The dataset was classified into two main categories, namely conservative and non-conservative water masses, with

non-conservative dataset further divided into 15 sub-categories to clarify and contrast with the more robust relationship between Kd(320nm/780nm) and acdom(440) for conservative water masses. The worldwide field dataset of downwelling irradiance (Ed) profiles collected with advanced optical instrumentation of C-OPS and C-PrOPS and the spectrophotometer-measured acdom for corresponding water samples presented in this study are valuable to investigate the CDOM contributions to Kd variations in several important marine ecosystems. However, the authors spend too much effort on data classification with text descriptions but did not present any visualized information of these valuable datasets, which are more interesting to the general readership. Further, it appears that the Methods section needs to be better organized as it is hard to follow. For Results section, discussion contents accounted for large portion of each sub-section, and some of the discussions appear speculative as it did not go deep enough to illustrate the real optical mechanism behind the relatively worse Kd ratioacdom(440) correlations. The overall impression about the article is positive, however, I would suggest the authors to reorganize the contents and work through the paper once more to revise the text and figures, making a worthy publication of such a valuable dataset.

Below are some specific comments: 1. Abstract, Page 1, line 18, 'e.g., increased vertical resolution to less than 1 mm...' Would like to see a better justification for this high vertical resolution even for turbid waters. Does the pressure sensor used in the C-OPS achieve such a resolution?

2. Introduction section: Some of content appear redundant, for example: (1) Page 2, line 27-28 and Page 3-4, the authors spend much effort describing the global perspective of sampling sites, which is a simple concept and can be properly combined and shortened. (2) It's better to combine the descriptions on the algorithms developed in Hooker et al. (2013) in Page 2, line 1-9 and Page 3, line 11-18 to avoid repetition and make it more clear to readers. (3) Page 2, line 17-18, looks not that accurate as there are many new sensors with high spatial resolution that have been used to study inland
and estuarine waters (e.g., Sentinel 2-MSI, Sentinel 3A/B-OLCI and Landsat 8-OLI), the authors may want to check on that and add some references. (4) It's better to move the summarized importance of acdom(440) in Page 2, line 19-25 to the first paragraph after line 3 "....by Nelson and Siegel, 2013"; it would be more coherent in content.

3. I would like to suggest the authors to reorganize Data and Methods section and again shorten the descriptions on global perspective sampling sites to avoid repetition. (1) Page 7, line 9, please use detailed number to replace "almost all". (2) Page 8, line 13, please add time period of your field dataset. (3) Page 11, line 9, please add reference after "...converted to absorption coefficient". (4) In section 2.5, Data Subcategories, can you provide more quantitative information of some of the criteria used for subcategories, such as, what's the Chlorophyll a value used to define algal bloom? what's the dominant species of HAB? You may want to mention this information as some of the HABs, like red tide species and cyanobacteria blooms display totally different optical properties, and further distinct Kd spectra; in addition, the authors mentioned that a sample sometimes satisfied more than one subcategory; so can you explain more about why classify data in this way which uses little quantitative information since the authors already utilized K-mean classification of Kd spectra, which make more sense. In the Results, the authors may want to compare Kd ratio with acdom(440) for each K-mean classified cluster to see if the correlations can be improved for each group.

4. In Results section, I would strongly recommend the authors add one more section to display some of interesting dataset (e.g., Kd and/or Ed spectra) collected in conservative and non-conservative water masses, such as, Hypersaline Lakes, river mouth, HAB.

5. In Results section, the authors displayed the correlations between Kd ratio and acdom(440) for different categories, however, they did not evaluate the performance of the algorithms, it's better to keep some dataset for validating their algorithm and study the errors and uncertainties using statistic parameters like RMSE, ARE, R2.
6. Page16, line3-7, the authors mentioned better correlation for hypersaline or alkaline lakes compared to the overfilled lakes, and explained turbidity could be the possible disturbance. More information should be provided, such as, how the turbid water modified the spectrum of Rrs, Ed and further Kd. What type of sediment, like mud, clay or silty increase the turbidity? Also, what's the "atypical constituents" in line 7, and can you add a reference here to support? How does this constituent influence the Kd spectra? Same thing for section 3.2, explain more about the influence of sediment-resuspension on high turbidity and on the variations of Kd.

7. Page 17, line 19-23, what's the atypical algal bloom, please add more information here.

8. Page 18, line 21-22, the author mentioned UV attenuation, which is likely due to production of UV-absorbing pigments (e.g., Mycosporine-like Amino Acids (MAAs)) by phytoplankton in response to UV stress. Please add more information and a reference here. Also, the authors may want to add information of the dominant species of algal bloom, because there are some species that can also strongly modify the spectrum in 700-800 nm range, like Trichodesmium.

9. The description on NASA NOMAD data should be moved to Data and Method section.

10. In section 3.7, some of contents relevant to method of K-mean classification seems to fit better in Methods, section 2. Further, it's better to move the whole section 3.7 up to the first sub-section in Results, which could help the general readers to better understand the algorithm performance (or nonperformance) of non-conservative waters.

BGD

---

## Referee Comment (RC2) · Anonymous Referee #2 · 21 Aug 2019

Overview

The manuscript is a fascinating and fun read. The authors present a robust dataset and thoroughly describe the methods used. However, significant revisions are suggested to more clearly link the author's subjective classification of environments to other literature discussing optical variability and the need to pre-screen for unique optical water types to effectively retrieve inherent optical properties from a given system. The authors would be well-served to present the full data set to display dynamic range across unique environments and graphically present the ability to measure radiometric variability at the millimeter scale – a fascinating accomplishment. A clearer link between

this capability and the decision to treat certain environments as anomalous is also warranted.

General comments

It seems the author's treated any data that did not conform to the algorithm as anomalous or atypical. Is this the best treatment of the dataset? This manuscript effectively serves as a more complete validation of the original end-member approach presented in Hooker et al. (2013). The goal of algorithms is to observe the environment "as is", so such a subjective treatment of any water that does not conform to anticipated algorithm output doesn't seem to be appropriate. Discussions of the "parent watermass" suggest that harbors, creek/river inputs, etc. are oddities; in fact, these spatial gradients are what we are trying to retrieve accurately with algorithms, and was a highlight of why the C-OPS was such an important instrument in the coastal zone in Hooker et al. (2013). The dataset is quite good and the algorithm useful. I think the author's would be best served by exploring the data, presenting the dynamic range (with associated categories, if necessary) and relate to algorithm performance. Discussion of potential improvements is also warranted, particularly considering that upcoming sensors are expected to have advanced spectral capabilities that, hopefully, will make band ratios less relevant for estimating a final product.

Presenting noise in measurements, and the ability to observe millimeter scale variability, would be quite useful as an additional figure. This was shown in Hooker et al. (2013) and was quite useful to fully understand the concept in that paper. Perhaps the existing figures showing performance across subjectively classified environments could be reduced into subplots of a single figure, allowing for fuller presentation of the dataset, including CDOM spectra and the ability to observe such fine scale variability in radiometry.

Was there any consideration of using CDOM spectral slope as a proxy of conservative versus non-conservative water masses?

A discussion of "anomalous" features is certainly warranted for radiometry measurements; however, it seems throughout that the author's measured "anomalous" conditions with the assumption that the Kd(320)/Kd(780) relationship observed for conservative waters holds to a universal truth across environments. This seems to be a gross simplification of the complexity of radiometry and a weakness of the current manuscript. As stated by the authors:

Lines 16-19, page 27: "The validation approach was based on the concept that water masses evolving conservatively (i.e., free from stressors that might cause anomalies to the natural range in the gradient of a constituent) are suitable for validating the original Hooker et al. (2013) inversion algorithm for deriving aCDOM(440) from Kd($\lambda$) spectral end members."

Effectively, observations that did not conform to the algorithm are considered "anomalous" and subjectively classified. Why did the authors not use an objective clustering approach to classify different environments, and consider an effective algorithm for each of these water types?

The introduction is fascinating and an interesting discussion. However, consider that the last ~1 page of text does not have a single citation. It seems much more relevant to tie this work more clearly into existing literature considering optical water variability due to different environments and optical complexity within a specific environment. This would leave much of the introduction intact while more clearly linking to how this builds upon past efforts, which it certainly does.

Specific Comments

Lines 16-22, page 7: These are helpful

All preceding paragraphs of this section (2.1) seem more suitable for supplementary material. I understand the motivation to show and describe improvements to the instrument from that shown in Hooker et al. (2013), but this takes valuable space away from

a more complete presentation of the dataset. Currently, all that is shown is the instrument, sampling locations and relationships with the empirical algorithm. Displaying the dynamic range of the data would be particularly useful.

Line 8, page 11: Why were only the Pacific Ocean and half of the Arctic Ocean samples baseline-corrected? Did use of 590-600 nm result in a significant offset for these spectra?

Lines 18-27, page 11: The section on western US coastal and inland water CDOM analysis is not clear. The two references to quantifying CDOM as the absorption coefficient at 440 nm and use of the Single Exponential Model make it seem that only CDOM at 440 nm was measured (there was no reference to quantifying CDOM at 440 nm for the other water samples). But, you also mention absorption spectra were measured – using Ultrapure water to dilute the signal? Were the samples optically thick and needed dilution? Please clarify this section, and to the extent possible, condense the sections on analysis of differently sourced water samples.

Section 2.5 The categorization of optical variability across water bodies is interesting, and the details are certainly relevant. However, it seems that the categorizations are rather subjective, and effectively used to explain outliers in the algorithm relationship. In the present state of the manuscript, this seems quite subjective. For a universal algorithm, it seems highly relevant to look for underlying means for deviations in the relationship that would aid in how the algorithm is applied. Effectively, you have categorized the environment that results in deviating optical properties that do not perform well within the algorithm; however, you haven't utilized the dataset to hypothesize on specific, only general, mechanisms (e.g., minerogenic content of particles and refractive index, spectrally different CDOM, dominance of phytoplankton absorption and scattering on Kd relationships rather than a relatively generic "sediment resuspension"). While quite difficult, the level of detail for the other sections seems to warrant this consideration. You have attempted to bypass this variability by using the end-member approach, targeting the wavelengths most and least influenced by aCDOM; other optical parameters significantly impacting the signal suggests a more detailed explanation, outside of categories, is warranted.

Lines 6-7, page 16: "...the new acreage is a source of atypical constituents, either in composition or concentration."

Really, this is the challenge of creating flexible, accurate algorithms that work across a variety of water types, either due to spatial, temporal or extreme event variability. It seems rather than addressing how to accurately estimate CDOM by modifying the algorithm, the authors highlight what is "abnormal" about these environments. There is certainly room for this, but I think the authors would be better served by focusing on how their algorithm could be adapted for these environments, rather than subjectively classifying environments where the algorithm does not perform well. The authors emphasize how the sensitivity of the instruments used detects these changes, but there is no analysis for how this increased sensitivity can be used to develop more capable algorithms.

Line 27-28, page 18: "For example, local wind conditions could elevate the values associated with a typical bloom into atypical concentrations"

Doesn't this call into question the purpose of classifications? These are conditions that will be observed, either through in situ or satellite observations. Could the algorithm be improved by factoring in wind conditions?

Lines 22-24, page 20: "As end members are brought spectrally closer together, the range of expression available to distinguish two similar but optically different water masses decreases."

Perhaps the mechanics of this could be explored and explained?

Lines 16-17, page 21: "Application of $\Lambda 412\ 670$ data to the corresponding algorithm in Fig. 8 results in 13 observations with negative (predicted) aCDOM(440) values, which are removed to leave 212 unique stations. This process demonstrates how endmember algorithms can be used to quality assure optical data in archives (Sect. 3.7)."

Lines 2-4, page 23: "With respect to the algorithm, the increased bias, variance, and 13 negative derived values obtained with NOMAD data (which is a small, quality controlled subset of the larger NASA SeaBASS archive) in clearer waters suggests the legacy data are degraded by sampling artifacts."

Is this an issue with the algorithm or the measurements?

Lines 14-15, page 23: "The data set established herein has an extensive number of observations directly suitable for validation exercises (Figs. 3 and 9) plus 15 subcategories (Sect. 2.5) of potentially (but not automatically) problematic water bodies (Figs. 4–7), with the latter determined subjectively."

The authors acknowledge that "problematic" water bodies were determined subjectively, and these observations do not agree well with the algorithm. Yet, don't these observations span natural environmental variability that can be observed? Outside of directly observing human structures (e.g., reflectance of a shipwreck visible from surface waters), why did the authors not consider how to retrieve valid aCDOM(440) values for these waters, and rather chose to assume the algorithm works very well and these waters are problematic? Further, it isn't clear how the algorithm can be used to determine whether legacy data is valid or not, as its performance is based on these subjective classifications, no?

Lines 7-8, page 24: "Consequently, a subcategorization scheme based on the optical measurements alone might be advantageous to the validation process, particularly for archival data."

This is assuming that CDOM should behave conservatively across water masses? It seems the very point the paper is making is that "abnormal" environments produce CDOM of a different spectral nature. This is important. Why have the authors not attempted to accurately estimate this variability?

Line 18-24, page 23: The author's reference clustering analysis, particularly fuzzy clustering presented by Moore et al. (various years) – why were subjective categorizations used rather than an objective approach such as that of Moore et al.? It would also be useful to reference this work earlier, perhaps in the introduction. Discussing the need for classifications to build effective algorithms could also be elaborated.

Lines 11-14, page 26: "The decrease in the percent composition of the validation quality data as a function of increasing class number (N1–N5) is an indicator of the difficulty of validating an algorithm within increasingly complex waters. The recurring contribution of a relatively small number of principal subjective subcategories to the gradient in optical complexity starting with N2 and then continuing for N3–N5 confirms the original subcategory approach has merit."

Conversely, the end member approach only performs well in bodies of water with little optical variability. It isn't clear why a subjective deconstruction of water bodies was used versus a fuzzy clustering approach where alternate relationships between Kd and aCDOM(440) were explored. Fuzzy clustering approaches use an objective classification scheme to separate out waters with the intention of providing a framework where different algorithms can be applied. Why was that not explored here?

Line 28, page 26: "evolving conservatively" – is this being used to represent anything that is a natural process within the water column, with no added optical constituents? Is photo- and microbial degradation of CDOM considered a conservative process? Perhaps a clearer discussion of Case 1 and Case 2 waters and how that classification relates to the classification used here would clarify this?

---

## Author Comment (AC1)

Dear Sirs:

Thank you for the opportunity to participate in the *Biogeosciences* review process. The paper for which I am the corresponding author is titled, "A Global End-Member Approach to Derive $a_{\mathrm{CDOM}}(440)$ from Near-Surface Optical Measurements" (BG-2019-259) by Stanford Hooker, Atsushi Matsuoka, Raphael Kudela, Youhei Yamashita, Koji Suzuki, and Henry Houskeeper. In the material presented below, page and line numbers are abbreviated as capital single letters, e.g., P2 L1–5 refers to page 2 lines 1 through 5.

**Reviewer 1 Comment 1:** The reviewer believes a better justification is needed for the high vertical resolution even for turbid waters and questions if the pressure sensor used in the C-OPS achieve such a resolution.
**Authors Response 1:** The accuracy of the pressure transducers used in this study have a depth resolution, in terms of precision, of 0.03–0.08 mm in all water masses. The manuscript will be modified by adding the precision values to the "less than 1 mm" vertical resolution statements in the abstract and P6 L25.

**Reviewer 1 Comment 2:** The reviewer states some content is redundant and believes the authors spend much effort describing the global perspective of sampling sites (P2 L27–28 and P3–4), which the reviewer believes is a simple concept that can be properly combined and shortened.
**Authors Response 2:** While the authors agree that maintaining brevity is crucial, defining the usage of a "global perspective" is necessary given the unique diversity of water masses sampled in this study, and because the two most common definitions are based on spatial extent and dynamic range. For example, if an algorithm is only accurate in the open ocean, such an algorithm would apply to the vast majority of the pixels in a worldwide remote sensing image and, thus, could be considered *global* even though it provides degraded—perhaps useless—information in the coastal zone and inland waters. The authors set out to create an algorithm that could be applied to the open ocean, coastal zone, and inland waters with equal efficacy, i.e., an algorithm that was arguably global in its application in terms of the dynamic range of water masses rather than the spatial extent of water masses.

Not wanting to belabor possible inadequacies in prior word usage, perhaps based primarily on spatial extent, the authors kept this topic short by *not* introducing *regional* and *universal* perspectives, which could be considered applicable to algorithm development. Reviewer 2, however, provided comments regarding the universal application of the algorithm, wherein a purported goal of an algorithm is to observe the environment "as is." Consequently, this material will be modified in the manuscript in keeping with the Authors Response 24 to Reviewer 2 while maintaining brevity to the greatest extent practicable.

**Reviewer 1 Comment 3:** The reviewer suggests combining the descriptions on the algorithms developed in Hooker et al. (2013) in P2 L1–9 and P3 L11–18 to avoid repetition and make it more clear to readers.
**Authors Response 3:** The authors agree and will move the applicable material presented in P3 to P2 within the manuscript.

**Reviewer 1 Comment 4:** The reviewer questions whether the statement that "this study includes coastal and inland ecosystems that are typically too small to be studied using common remote sensing platforms" (P2 L17–18) is accurate, because the reviewer asserts "there are many new sensors with high spatial resolution that have been used to study inland and estuarine waters" (e.g., MSI, OLCI, and OLI).
**Authors Response 4:** The intention of the authors was not to dismiss the fairly large body of work on coastal waters using MSI and OLI (among others), but rather to point out that many of the field sites included in this analysis are still not effectively sampled by those platforms, because three contiguous water pixels are required to ensure that imagery is free from edge effects (i.e., stray light). This limits sensors such as MSI, OLI, and OLCI to water bodies with spatial scales exceeding 30, 90, and 900 m, and this study sampled water masses that were smaller in one or more of these dimensions (e.g., rivers). While the aforementioned sensors have been used extensively for coastal and inland waters, e.g., Palmer et al. (2015) and Mouw et al. (2015), high spatial resolution sensors typically have reduced spectral resolution or range, and existing methods for characterizing $a_{\mathrm{CDOM}}(\lambda)$ have been inadequate even within relatively large, lacustrine water bodies (Kutser et al. 2015). The manuscript will be modified to briefly clarify this point (P2 L17–18).

**Reviewer 1 Comment 5:** The reviewer recommends combining the descriptions on the algorithms developed in Hooker et al. (2013) in P2 L1–9 and P3 L11–18 to avoid repetition and make it more clear to readers.

**Authors Response 5:** The authors agree and will modify the manuscript by moving the applicable material presented in P3 to P2.

**Reviewer 1 Comment 6:** The reviewer believes it is better to move the summarized importance of $a_{\text{CDOM}}(440)$ in P2 L19–25 to immediately after the sentence ending in P2 L3.

**Authors Response 6:** The authors agree and will modify the manuscript accordingly.

**Reviewer 1 Comment 7:** The reviewer suggests shortening the descriptions on global perspective sampling sites to avoid repetition.

**Authors Response 7:** The authors agree and will modify the manuscript (P7–8) accordingly.

**Reviewer 1 Comment 8:** The reviewer suggests using a detailed number to replace the phrase "almost all" in P7 L9.

**Authors Response 8:** The authors agree and will modify the manuscript by replacing "Almost all" with "Approximately 98%" (P7 L9).

**Reviewer 1 Comment 9:** The reviewer suggests adding the time period over which the field dataset was collected (P8 L13).

**Authors Response 9:** The authors agree and will modify the manuscript by adding "data collection spanned 29 April 2013 to 25 January 2017" (P8 L13)).

**Reviewer 1 Comment 10:** The reviewer suggests adding a reference regarding the conversion to absorption coefficient (P11 L9).

**Authors Response 10:** The authors agree and will modify the manuscript by adding the Green and Blough (1994) citation in P11 L9.

**Reviewer 1 Comment 11:** The reviewer suggests more quantitative information of some of the criteria used for subcategories is needed in Sect. 2.5, such as, what's the chlorophyll $a$ value used to define algal bloom and what's the dominant species of HAB? The reviewer notes that the authors may want to mention this information as some of the HABs, like red tide species and cyanobacteria blooms display totally different optical properties, and further distinct Kd spectra.

**Authors Response 11:** Defining a HAB event is somewhat fuzzy. For example, Smayda (1997) (correctly) points out that an absolute chlorophyll concentration has little to do with how so-called "blooms" are labeled, while HABs can occur at very low chlorophyll if they are dominated by noxious or toxic algae. The authors have followed typical convention for identification of a HAB as an event dominated by a known HAB organism, or causing a deleterious effect on humans or the environment. The authors will expand the manuscript slightly (P12 L26–27) to indicate that a HAB is subjectively determined with these objective criteria.

**Reviewer 1 Comment 12:** The reviewer notes that the authors mentioned that a sample sometimes satisfied more than one subcategory, and then requests additional information as to why the authors classified data in this way which uses little quantitative information since the authors already utilized K-mean classification of Kd spectra, which make more sense.

**Authors Response 12:** The subjective classification scheme makes use of relevant factual information known about a water mass, which is not always available in algorithm development or validation, thus the inclusion of both objective and subjective approaches in this study. The text introducing the subjective subcategories (P23 L14–21) will be modified to clarify the underlying characteristics of each subcategory are based on factual and quantitative observations essential to the qualities and attributes of each one, i.e., the subcategories are not arbitrary or lacking in substance.

As presented at the start of Sect. 2.5, the subjective subcategories were based on sampling information directly associated with optical properties, which would otherwise not be known. The subcategories also provided a running inventory of the types of water masses that were incrementally sampled to ensure a global sampling with a representative number of each type of water mass to the extent practicable. This approach was effective as proved by the amount of data in each FCM class (Table 2): $N_1$ 244, $N_2$ 263, $N_3$ 305, $N_4$ 265, and $N_5$ 94. The amounts are rather balanced except for the most extreme observations in the $N_5$ class, which were the most difficult to obtain (extreme water masses are not common).

The fewer samples in the extreme $N_5$ class do not negate the overall applicability of the subcategorization scheme. For each FCM class, the cumulative percentage of the dynamic range in $K_d$ end members and $a_{\mathrm{CDOM}}(440)$ is, respectively, as follows: $N_1$ 2.3% and 3.4%; $N_2$ 8.4% and 8.9%; $N_3$ 22.7% and 36.4%; $N_4$ 82.9% and 100.0%; and $N_5$ 100.0% and 100.0%. These data reveal that the $a_{\mathrm{CDOM}}(440)$ dynamic range is completely established with the addition of the $N_4$ class, which provides the largest extension of the optical and biogeochemical dynamic ranges. The $N_1$ (case-1) class makes the smallest contribution to the dynamic ranges, although it arguably accounts for most of the pixels in a global CDOM image, and the $N_5$ class only extends the optical dynamic range. Consequently, the global algorithm is comprehensively established by the $N_1$–$N_4$ combined classes and the $N_5$ class primarily contributes variance to the algorithm, i.e., it does not expand the optical versus biogeochemical relationship provided by classes $N_1$–$N_4$.

An algorithm based on all the data in classes $N_1$–$N_4$ yields a linear fit of $y = 0.2317x - 0.0053$, the RMSE is 5.3%, and the slope is to within 9.3% of the original value presented by Hooker et al. (2013). This result is significantly similar (the slopes agree to within 2.7%) to the subjective results discussed as the "fourth more comprehensive data set" in Sect. 4 (P29 L19–23), wherein $y = 0.2379x - 0.0049$, the RMSE is 6.2%, and the slope is to within 6.9% of the original value presented by Hooker et al. (2013). Consequently, the robustness of the algorithm is directly supported by the combination of subjective and objective classifications, with the latter using fuzzy c-means (FCM). It is important to remember that the FCM approach is different from a hard or crisp classification, such as $k$-means which was not used.

The authors believe the subjective-objective approach that was used facilitates an appropriate interpretation of the results, because the subjective approach reveals cause-and-effect relationships (e.g., the importance of water masses subjected to resuspension effects in higher class numbers as shown in Table 2), and the objective approach provides an unbiased strictly quantitative confirmation. Table 2 will be expanded a little bit to present the extent of the dynamic range analysis and summarize its importance. In addition, the manuscript will be modified to succinctly present the $N_1$–$N_4$ algorithm results in companion with the "fourth more comprehensive data set" as a fifth comprehensive data set in Sect. 4 (P29 after L23).

**Reviewer 1 Comment 13:** The reviewer suggests a comparison of the Kd ratio with aCDOM(440) for each K-mean classified cluster to see if the correlations can be improved for each group.

**Authors Response 13:** The authors agree that comparisons within each FCM category would be valuable, and the extra material will be succinctly presented in the manuscript Sect. 4 and linked with the new material to be added as part of Authors Response 41 (which will appear at the end of Sect. 3.7 P27 L13, i.e., right before Sect. 4 (P27 L14).

**Reviewer 1 Comment 14:** The reviewer strongly recommends the authors add one more section in the Results section to display some of interesting dataset (e.g., Kd and/or Ed spectra) collected in conservative and non-conservative water masses, such as, Hypersaline Lakes, river mouth, HAB.

**Authors Response 14:** While the authors appreciate the interest in $K_d$ and $E_d$ spectra, they must respectfully decline for the following reasons: a) the objective of the paper is to produce a simple reliable algorithm using end-member analyses and not to describe the optical properties of a large number of water masses; b) the study contains so many observations that individual spectra overlap each other significantly and create a continuum of lines from the pure water limit to the most turbid $N_5$ water body (White Lake), which are difficult to discern and which do not provide any additional information beyond what is already presented in the manuscript; c) if multiple panels are used to magnify certain parts of the dynamic range, the amount of required text significantly lengthens the manuscript; and d) there is persistent pressure from both reviewers to produce a more compact manuscript. Consequently, no new figures will be added to the manuscript and the only new sections are the result of moving material at the request of the reviewers (see Authors Responses 19 and 20).

**Reviewer 1 Comment 15:** The reviewer notes the authors displayed the correlations between Kd ratio and aCDOM(440) for different categories, however, they did not evaluate the performance of the algorithms, it's better to keep some dataset for validating their algorithm and study the errors and uncertainties using statistic parameters like RMSE, ARE, R2.

**Authors Response 15:** The approach of the original Hooker et al. (2013) manuscript, as well as this study, already adhere to the Reviewer's comment as follows: a) Hooker et al. (2013) used a separate validation subset collected in significantly different geographical areas, which was used to confirm the efficacy of the derived algorithm and quantify performance parameters (e.g., RMSE, $R^2$, etc.); b) the entire dataset used for this study is a distinct and separate set of observations from significantly different geographical areas that is used to validate the original algorithm; c) the categorization scheme partitions the observations into conservative and non-conservative water masses and the influence of adding in the non-conservative fraction is quantified in terms of the same performance parameters, so there is a progression of algorithm validations based on increasingly larger data subsets; and d) NOMAD data were the only independent validation subset that could be found, but the poorer spectral and geographical diversity limits their utility. To clarify this aspect of validation, the manuscript will be slightly modified in the new NOMAD Sect. 2.6 (see Authors Response 19) to remind the reader that the validation process used herein adheres to the concept of having a separate dataset to estimate the uncertainties using statistical parameters and the use of the NOMAD data is an extension of that philosophy.

**Reviewer 1 Comment 16:** The reviewer notes the authors mentioned better correlation for hypersaline or alkaline lakes compared to the overfilled lakes, and explained turbidity could be the possible disturbance (P16 L3–7), and suggested the following: a) more information should be provided, such as, how the turbid water modified the spectrum of Rrs, Ed and further Kd; b) what type of sediment, like mud, clay or silty increased the turbidity; and c) identification of the "atypical constituents" in L7, perhaps with a supporting reference; d) an explanation of how this constituent influences the Kd spectra; and e) for section 3.2, an explanation about the influence of sediment-resuspension on high turbidity and on the variations of Kd.

**Authors Response 16:** The referenced pages deal with lacustrine subcategories, wherein the manuscript text notes that refilled lakes frequently exhibit larger anomalies with respect to the algorithm than hypersaline or alkaline lakes, especially in terms of turbidity as determined by the $K_d$ ratio. The "atypical constituents" introduced to a water mass when it is overfilled is a generalized phenomenon, wherein land that is subjected to other purposes (e.g., agricultural and anthropogenic activities associated with grazing, farming, vehicular traffic, etc.) will provide one or more constituents to the water mass when the lake overfills that are not typical of what is in the water mass prior to overfilling, because these activities are not possible in the water mass. The manuscript will be slightly modified to make this point clearer in P16 L6–7.

In regards to the list of requested clarifications the principal two problems are as follows: a) the manuscript does not rely on $R_{rs}(\lambda)$ or $E_d(\lambda)$, so these variables are not a part of this study; and b) determining how turbidity modified the $K_d(\lambda)$ spectrum, determining what type of sediment increased the turbidity, identifying "atypical constituents" introduced into a lake when it overfills, explaining how the introduction of an "atypical constituent" influences $K_d$ spectra, and explaining more about the influence of sediment resuspension on high turbidity and on the variations of $K_d(\lambda)$ all require baseline data for the subject water masses prior to modification, which are not available. Although this entire line of inquiry indicates interest in the work that the authors performed, it is not objectively focused on algorithm validation. Instead, it involves scientific pursuits that either cannot be answered, because of a lack of baseline data, or are outside the scope of the material presented (e.g., $R_{rs}$ and $E_d$ spectra). Consequently, the manuscript will not be modified for these itemized comments.

**Reviewer 1 Comment 17:** The reviewer requests more information regarding the atypical algal bloom (P17 L19–23).

**Authors Response 17:** The language regarding an "atypical bloom" was meant to describe a generic case of a water mass wherein there was unusually high biomass of (typically) a single species of algae. An example was given of physical forcing (wind and waves) accumulating unusually high biomass on one side of a lake, although similar phenomena occur in the coastal ocean, whereby a combination of algal growth, vertical migration (behavior), and physical aggregation can on occasion result in dinoflagellate blooms reaching

more than $1,000\,\mu\mathrm{g\,L}^{-1}$ of chlorophyll in Monterey Bay, which is about 10 times higher than (already high-biomass) red tide events that are not physically aggregated. To put this into context, the manuscript will be modified slightly by adding the Kudela et al. (2015) reference to P17 L23. The Kudela et al. (2015) study documented concentrations of chlorophyll in excess of $2,000\,\mu\mathrm{g\,L}^{-1}$ at Pinto Lake, one of the water bodies included in Fig. 6 as anomalous.

**Reviewer 1 Comment 18:** The reviewer notes that the authors mentioned UV attenuation (P18 L21–22), which is likely due to production of UV-absorbing pigments (e.g., Mycosporine-like Amino Acids (MAAs)) by phytoplankton in response to UV stress, and suggested more information and a reference. The reviewer also suggested the authors may want to add information of the dominant species of algal bloom, because there are some species that can also strongly modify the spectrum in 700-800 nm range, like Trichodesmium.

**Authors Response 18:** The reviewer is correct, some of the blooms (marine dinoflagellates in particular) are associated with MAA-like compounds that strongly impact UV absorption. The HAB events identified in Fig. 6 as Monterey Bay were dominated by *Cochlodinium* and *Akashiwo*. For *Akashiwo* in particular, MAA-like compounds are a diagnostic indicator of the presence of foam-producing substances (Jessup et al. 2009), while Kwon et al. (2018) demonstrate significant increases in FDOM and DOC in *Cochlodinium* blooms.

Unfortunately, even if phytoplankton absorption coefficient spectra were available, it would still be difficult to unequivocally ascertain the presence of MAAs, because no valid beta factor that can rigorously be applied for the UV spectral domain is presently available. To avoid speculation, the authors decided not to expand the discussion about MAAs in the text, but the manuscript will be modified slightly with a small revision to include the aforementioned references in P18 L22.

**Reviewer 1 Comment 19:** The reviewer suggests the description on NASA NOMAD data should be moved to Data and Method section.

**Authors Response 19:** The authors agree, so the manuscript will be modified by adding a new Sect. 2.6 using the appropriate material in Sect. 3.6.

**Reviewer 1 Comment 20:** In regards to Sect. 3.7, the reviewer suggests some of contents relevant to method of K-mean classification seems to fit better in Methods (Sect. 2) and proposes it's better to move the whole section 3.7 up to the first sub-section in Results, which could help the general readers to better understand the algorithm performance (or nonperformance) of non-conservative waters.

**Authors Response 20:** Recalling that this study does not use $k$-means classification (it uses FCM classification), the authors agree to modify the manuscript by moving some of the FCM contents in Sect. 3.7 to a new Sect. 2.7. The authors disagree with moving Sect. 3.7 to the first section in Results, because this would place the material being moved before the description of the subjective classification data and the material being moved references these data, so the subjective classification data must appear first, so there will be no modification of the manuscript for this part of the comment.

**Reviewer 2 Comment 1:** The reviewer asserts there is a need to a) clearly link the subjective classification of environments to other literature discussing optical variability, and b) pre-screen for unique optical water types to effectively retrieve inherent optical properties from a given system.

**Authors Response 21:** The algorithm development approach espoused in the manuscript classified the data subjectively, so more complex water masses could be added to the algorithm incrementally to provide quantitative assessments of how the more complex water masses influenced algorithm performance. No other purpose was espoused or documented in the manuscript. To clarify the purpose of the subjective approach, the text introducing the subjective categories (P23 L14–21) will be modified to make it clear that the "subcategories are used exclusively to assess algorithm performance as more complex water masses are included."

The second assertion requiring a need to pre-screen for unique optical water types to effectively retrieve IOPs from a given system is not scientifically objective, because the study makes no effort to do this; in fact, it seeks to accomplish the opposite, i.e., retrieve an in-water biogeochemical constituent, $a_{\mathrm{CDOM}}(440)$, from observations of the diffuse attenuation coefficient, $K_d(\lambda)$. No modifications of the manuscript are made for this comment. Consequently, no modifications will be made to the manuscript for this comment.

**Reviewer 2 Comment 2:** The reviewer states the authors would be well-served to present the full data set to display dynamic range across unique environments and graphically present the ability to measure radiometric variability at the millimeter scale—a fascinating accomplishment. A clearer link between this capability and the decision to treat certain environments as anomalous is also warranted.

**Authors Response 22:** Figures 3–8 and 10 present all of the data used in the study while identifying all of the unique environments that were sampled with almost all of them labeled. From the perspective of the capabilities of a COTS pressure transducer, the ability to measure radiometric variability at the millimeter scale is well established. What makes the accomplishment unique is the use of small digital thrusters to maneuver the optical backplane while maintaining the planar orientation of the radiometer apertures. The latter ensures very little data, and typically no data in inland waters where vertical resolution is critical, are lost because the vertical tilt of the instruments exceed $5°$. Consequently, the primary reason radiometric variability is measured at the millimeter scale is because of the the precision of the pressure transducer, the high data rate of C-OPS microradiometers, the slow sinking rate near the surface from the *hydrobaric* buoyancy chamber, plus the planar stability from the C-PrOPS accessory. The former is now included in the manuscript as part of the Authors Response to Reviewer 1 Comment 1, the latter is documented in the manuscript as part of describing the efficiencies of thruster-assisted profiling (P7 L16–22), and the data rate with Morrow et al. (2010) reference will be added as a small addition to the manuscript at P7 L16–22. The hydrobaric buoyancy chamber is documented in Fig. 1. A graphical depiction is not deemed necessary, because it will lengthen the manuscript without adding any value beyond what is already reported in the manuscript, so the manuscript will not be further modified.

**Reviewer 2 Comment 3:** The reviewer posits that it seems the author's treated any data that did not conform to the algorithm as anomalous or atypical.

**Authors Response 23:** The definition of conservative water masses (P3 L28–29) and the follow-on definition of what is considered an anomalous condition (P3 L29 to P4 L1–3) were used to establish the subjective subcategories. The merits of classifying the data using subjective criteria is easily discerned by comparing Figs. 3 and 5, i.e., conservative water masses versus a subset of non-conservative water bodies. Rivers were arguably one of the most difficult ecosystems sampled, because they are by definition rather shallow and the moving water makes profiling challenging. Nonetheless, Fig. 3 contains many examples of the inland portion of rivers, which are characterized as conservative water masses despite the sampling difficulties. The Sacramento River at flood stage, however, was categorized as a resuspension water body because resuspended material was visible, and its similarity with other resuspended water masses in respect to the algorithm is apparent in Fig. 5. River mouths (which represent a mixing of water masses) also group together, although differently than rivers or resuspension in respect to the algorithm. The merits of classifying the data was also proven by using an objective FCM scheme to show the data naturally classify into groups, and five classes were identified.

The fact that the various groups of data have dissimilar relationships with the algorithm does not mean they were treated differently. Ultimately, the manuscript shows in Sect. 4 P29 L19–23 that if all the data except extreme lacustrine water bodies (e.g., the White Lake data had estimated values in the UV domain, Bear Lake is a unique scattering anomaly created by calcium carbonate particles, etc.) are used to create a fourth data set with 1,086 observations—i.e., almost 90% of the 1,230 maximum and 93% of the data used in Table 2 to create the five objective FCM classifications—the linear fit of the fourth more comprehensive data set is $y = 0.2379x - 0.0049$, the RMSE is 6.2%, and the new slope is to within 6.9% of the original value presented by Hooker et al. (2013).

To prevent a similar erroneous understanding, the manuscript will be modified immediately after the sentence ending on P4 L25 by adding the following: "Ultimately, all subcategories are incrementally added to the algorithm evaluation process to assess performance as a function of increasing water mass complexity." Furthermore, to clarify that the subjective approach correctly categorizes the data, the manuscript will be modified slightly in P16 after L19 by adding a new sentence: "The dissimilar expression of the flooded Sacramento River with respect to the inland riverine data in Fig. 3 not in flood conditions (i.e., as conservative water masses), shows the subjective classification approach has merit."

**Reviewer 2 Comment 4:** The reviewer states the goal of algorithms is to observe the environment "as is",

so such a subjective treatment of any water that does not conform to anticipated algorithm output doesn't seem to be appropriate.

**Authors Response 24:** All of the water masses sampled in this study were observed "as is" and ultimately 90% or more of the data were used to evaluate algorithm performance. The small amount of data that were not included in the fourth algorithm data set were properly excluded as detailed in the manuscript (e.g., the White Lake data had estimated values in the UV domain, Bear Lake is a unique scattering anomaly created by calcium carbonate particles, etc.).

More importantly, the analysis presented was not aimed at removing "anomalous" water masses that did not conform to the proposed end-member analysis (EMA), but rather to identify what situations lead to departures from the algorithm. The authors agree that an algorithm based on first principals should be able to fit all naturally (and artificial) occurring samples, but the authors also never claimed that there is a fundamental law or physical concept that would fit such an algorithm. Rather, the authors point out that "anomalous" water bodies that deviate from the predicted relationship can generally be explained and subjectively or objectively classified based on their optical properties.

The anomalous points may not be ideal for validation of the algorithm but are nonetheless incrementally included in evaluating algorithm performance, as presented in Sect. 4. Within the calibration validation research (CVR) paradigm, these same points would certainly be in-scope for research, and the EMA approach would help to define the research (i.e., determining what characteristics about the water masses makes them anomalous when using the EMA approach). It is important to recall that in terms of surface area, the "anomalous" water masses constitute a tiny fraction of the total aquatic area of worldwide ecosystems, thereby confirming that for the vast majority of cases the approach will work well. The manuscript will be modified at the introduction of the subjective subcategories in Sect. 2.5 (P13 L18) to clarify the "anomalous" water masses are a tiny fraction of the total aquatic area of worldwide ecosystems.

**Reviewer 2 Comment 5:** The reviewer asserts discussions of the "parent water mass" suggest that harbors, creek/river inputs, etc. are oddities; in fact, these spatial gradients are what we are trying to retrieve accurately with algorithms, and was a highlight of why the C-OPS was such an important instrument in the coastal zone in Hooker et al. (2013).

**Authors Response 25:** While the authors agree that the C-OPS instrument suite has improved characterization of coastal zone features such as spatial gradients, the parent water mass discussion is solely applied to demonstrate sensitivity, and all parent water mass modifier data were included in algorithm performance evaluations. In addition, no water masses described in the manuscript are considered to be oddities by the authors. The use of parent water mass modifiers was explained in the manuscript as "a localized alteration of water properties, e.g., a creek inflow into a lake, and demonstrates the sensitivity of the methods used herein to distinguish small changes." All the parent water mass modifier data were used to demonstrate sensitivity and all were included in algorithm performance evaluations. Furthermore, the spatial gradients associated with 90% or more of the water masses were included in evaluating algorithm performance. The small amount of data that were not included in the fourth algorithm data set were properly excluded as detailed in the manuscript (e.g., the White Lake data had estimated values in the UV domain, Bear Lake is a unique scattering anomaly created by calcium carbonate particles, etc.). The authors believe the present form of the manuscript properly presents all forms of sensitivity arguments, so no modifications will be made.

**Reviewer 2 Comment 6:** The reviewer thinks the author's would be best served by exploring the data, presenting the dynamic range (with associated categories, if necessary) and relate to algorithm performance.

**Authors Response 26:** The authors agree that presenting dynamic range and relating associated categories to algorithm performance are important, and believe that Figs. 3–8 and 10 with accompanying text succinctly and thoroughly addresses these requests. In addition, Sect. 4 already explores algorithm performance as a function of increasing water mass complexity, so no modifications to the manuscript were deemed necessary. Consequently, no modifications to the manuscript will be made.

**Reviewer 2 Comment 7:** The reviewer suggests discussion of potential improvements is also warranted, particularly considering that upcoming sensors are expected to have advanced spectral capabilities that, hopefully, will make band ratios less relevant for estimating a final product.

**Authors Response 27:** The authors have expanded on this point at the end of Sect. 4. It is important to note that while some remote sensors (e.g., PACE) will provide much improved spectral capabilities and the ability to employ other algorithm approaches (spectral shape, semi-analytical inversion, etc.) other sensors (e.g., MSI, OLI, OLCI, etc.) are still multispectral, while SBG has yet to be fully defined. Even with spectrometer-based systems (PACE, possibly SBG) the EMA approach provides a simple and independent way to assess data quality and algorithm performance for more sophisticated algorithms, while the ability to estimate CDOM at the millimeter depth scale with C-OPS, or more generally to conduct the same measurements with a two-channel system, provides an ability to generate vast quantities of data compared to traditional optical measurements, improving both validation and research for coupled remote sensing and *in situ* studies.

The manuscript will be modified after P31 L29 by adding the following: "While planned high spectral resolution sensors, such as the Plankton, Aerosol, Cloud, ocean Ecosystem (PACE) and Surface Biology and Geology (SBG) missions, may support more sophisticated retrievals of parameters such as CDOM, the simplified approach provided by end-member analysis can in principal be used with both legacy and next-generation sensors, thereby providing continuity in space and time, as well as a capability to generate high-quality in-water data with a simplified measurement approach (assuming rigorous adherence to the sampling protocols)."

**Reviewer 2 Comment 8:** The reviewer suggests presenting noise in measurements, and the ability to observe millimeter scale variability, would be quite useful as an additional figure.

**Authors Response 28:** Appropriate noise estimates already appear in the manuscript either in terms of method performance in Sect. 4 (P30 L27 to P31 L5) and in Figs. 3–8, which include all optical casts associated with each biogeochemical measurement, so the noise is represented graphically. The sensitivity of the methods used is presented graphically as part of showing the parent water mass modifiers in Fig. 7. A separate graphical depiction of millimeter scale variability is unnecessary, because it will lengthen the manuscript without adding any value beyond the metrics already reported in the manuscript. Consequently, no modifications to the manuscript were deemed necessary.

**Reviewer 2 Comment 9:** The reviewer suggests that the existing figures showing performance across subjectively classified environments could be reduced into subplots of a single figure, allowing for fuller presentation of the dataset, including CDOM spectra and the ability to observe such fine scale variability in radiometry.

**Authors Response 29:** Figures 3–8 and 10 present all of the data used in the study while identifying all of the unique environments that were sampled with almost all of them labeled. CDOM spectra were not used in the study, so their presentation is unnecessary. The ability to observe fine-scale variability is addressed in Authors Response 22.

**Reviewer 2 Comment 10:** The reviewer questions whether there was any consideration of using CDOM spectral slope as a proxy of conservative versus non-conservative water masses.

**Authors Response 30:** Although an interesting idea, the authors did not consider any alternative to identifying conservative versus non-conservative water masses for a few reasons. First, CDOM was measured separately in the various laboratories, and spectral slope was also determined using slightly different methodologies, so an analysis as to whether that had an impact would be needed. Second, our primary focus was on using the bio-optical data, with subjective or objective (clustering) criteria, and there is no simple way to derive the spectral slope from the EMA approach (it would have to be tested and validated independently of the existing methodology). Third, some authors have suggested that spectral slope does not vary as widely as is assumed, and that much of the difference is related to analytical methodology (which brings us back to the first point), e.g., Twardowski et al. 2004. Consequently, no modifications to the manuscript were considered necessary.

**Reviewer 2 Comment 11:** The reviewer states that a discussion of "anomalous" features is certainly warranted for radiometry measurements; however, it seems throughout that the author's measured "anomalous" conditions with the assumption that the Kd(320)/Kd(780) relationship observed for conservative waters

holds to a universal truth across environments. This seems to be a gross simplification of the complexity of radiometry and a weakness of the current manuscript. As stated by the authors: P27 L16-19: "The validation approach was based on the concept that water masses evolving conservatively (i.e., free from stressors that might cause anomalies to the natural range in the gradient of a constituent) are suitable for validating the original Hooker et al. (2013) inversion algorithm for deriving aCDOM(440) from Kd($\breve{\lambda}$) spectral end members." Effectively, observations that did not conform to the algorithm are considered "anomalous" and subjectively classified.

**Authors Response 31:** Nowhere in the manuscript do the authors state or imply that the $K_d(320)/K_d(780)$ relationship observed for conservative waters holds to a universal truth across environments. Instead, the authors established a plausible hypothesis for establishing a starting point for the validation process, evaluated algorithm performance while incrementally adding water masses of increasing optical complexity, and ultimately established the original algorithm had an accuracy of approximately 6% while using 90% or more of the water masses spanning three decades of optical and biogeochemical dynamic range. The small amount of data that were not included in the fourth algorithm data set were properly excluded as detailed in the manuscript (e.g., the White Lake data had estimated values in the UV domain, Bear Lake is a unique scattering anomaly created by calcium carbonate particles, etc.).

The authors note that the reviewer's interpretation (that observations that did not conform to the algorithm are considered "anomalous" and subjectively classified) is not in agreement with the methodology performed in this study. In fact, the subjective classification took place prior to sampling a particular site for a particular water mass to ensure that the process was unbiased and free of any influence from the actual observations. The manuscript will be slightly modified by clarifying that the subjective classification took place prior to sampling a particular site for a particular water mass in P12 L5–7.

**Reviewer 2 Comment 12:** The reviewer questions why the authors did not use an objective clustering approach to classify different environments, and consider an effective algorithm for each of these water types.

**Authors Response 32:** The goal of the study was to produce a global algorithm that would be evaluated by adding in incrementally more complex waters while quantifying the effect on the algorithm. At no point was a series of regional or classification algorithms considered. Within this context the "anomalous" water masses are atypical compared to what would occur with conservative mixing of defined end members and have an unknown additional optical complexity. As noted in Authors Response 21, the authors do not state that the algorithm is based on some *universal truth*, and did not mean to imply that water masses that do not fit the conservative concept are not natural in the broader sense.

The argument for subjectively classifying them is because a priori subjective information was used first that is not strictly derived from the optics. For example, if optical sampling is conducted near an ice edge, or in a drained and refilled lake, then it is logical to assume that this forcing has some impact on the optical properties, but a purely objective classification scheme may not partition those conditions, because there is not necessarily a unique optical property that is associated with those physical, biological, or chemical phenomena. So the point was to start with a subjective classification scheme and identify data that do not conform to the algorithm (anomalies), then to determine whether there is some clustering based on additional information that exposes where those observations fall when compared to the expected relationship, and then to evaluate the effect on algorithm performance as these optically more complex data are incrementally added to the algorithm.

The reviewer states that observations that did not conform are anomalous, and therefore subjectively classified. All data were first subjectively classified (prior to data collection in the field, per Authors Response 31) and then plotted against the algorithm. Some did not conform to the generalized relationship and they were evaluated with additional (non-optical) information to identify a proximate cause. An objective clustering would still highlight those as non-conforming; the subjective clustering attempts to provide a rational explanation for why they differ. Separate algorithms based on the objective classification were not developed, because the goal was to test a globally-applicable algorithm and not to develop a set of related but different regional algorithms. While the latter certainly could be done, it was not the primary focus of this study.

To ensure this question is addressed in the manuscript, the manuscript will be slightly modified in P12 L5–7 by clarifying that the algorithm validation process begins with the conservative water mass data and

algorithm performance is further quantified by incrementally adding more complex water masses from the subcategories to the evaluation data set.

**Reviewer 2 Comment 13:** The reviewer states the introduction is fascinating and an interesting discussion. However, consider that the last ¸1 page of text does not have a single citation. It seems much more relevant to tie this work more clearly into existing literature considering optical water variability due to different environments and optical complexity within a specific environment. This would leave much of the introduction intact while more clearly linking to how this builds upon past efforts, which it certainly does.

**Authors Response 33:** The authors agree and have added the following citations to provide the requested links: a) P3 L23 (Yapiyev et al. 2017); b) P3 L25 (Bodaker et al. 2010); c) P4 L10 (Lee and Hu 2006); d) P4 L15 (Morel 1974); e) P4 L24 (Guarch-Ribot and Butturini 2016 and Vazquez et al. 2011).

**Reviewer 2 Comment 14:** The reviewer states P7 Lines 16-22 are helpful and that all preceding paragraphs of this section (2.1) seem more suitable for supplementary material. The reviewer acknowledges the motivation to show and describe improvements to the instrument from that shown in Hooker et al. (2013), but the reviewer believes this takes valuable space away from a more complete presentation of the dataset. Currently, all that is shown is the instrument, sampling locations and relationships with the empirical algorithm. Displaying the dynamic range of the data would be particularly useful.

**Authors Response 34:** While the authors support the goal of increasing the brevity of the text, they believe that the information described is necessary given that knowledge of the technology to observe optical variability at the 1 mm scale is a) not widely known throughout the community of practice, and b) was central in enabling the accomplishments of this manuscript. In addition, the authors note that the manuscript already contains a complete presentation of the data used as well as the dynamic range of the data in Figs. 3–8 and 10. Consequently, no modifications will be made to the manuscript.

**Reviewer 2 Comment 15:** The reviewer questions why only the Pacific Ocean and half of the Arctic Ocean samples were baseline-corrected and wonders if use of 590-600 nm result in a significant offset for these spectra (P11 L8).

**Authors Response 35:** The other half of Arctic Ocean samples were also baseline-corrected with the mean value of $a_{\mathrm{CDOM}}(\lambda)$ between 683 and 687 nm according to a reference cited in the manuscript (Matsuoka et al. 2017). The effect of the three different laboratory methods (including the different wavelengths used for baseline correction) on $a_{\mathrm{CDOM}}(440)$ values were tested. As a result, it was found that the use of three different laboratory methods to determine $a_{\mathrm{CDOM}}(440)$ does not significantly influence the results presented in the manuscript. This indicates that use of 590–600 nm did not result in a significant offset for these spectra, as represented by $a_{\mathrm{CDOM}}(440)$, compared to the other methods. This issue was presented in Sect. 4 (P30 L20-29 to P31 L1-5). The manuscript will be briefly modified in P11 L8 to make it clear that the other half of the Arctic Ocean samples were baseline-corrected and also after P31 L5 to state the use of 590–600 nm did not result in a significant offset.

**Reviewer 2 Comment 16:** The reviewer says the section on western US coastal and inland water CDOM analysis is not clear (P11 L18–27), as follows: a) the two references to quantifying CDOM as the absorption coefficient at 440 nm and use of the Single Exponential Model make it seem that only CDOM at 440 nm was measured (there was no reference to quantifying CDOM at 440 nm for the other water samples); b) the authors, however, also mention absorption spectra were measured, perhaps using Ultrapure water to dilute the signal; and c) Were the samples optically thick and needed dilution? Please clarify this section, and to the extent possible, condense the sections on analysis of differently sourced water samples.

**Authors Response 36:** To clarify the description about CDOM analysis for western US coastal and inland water, the manuscript will be modified as follows: "For the western US coastal and inland waters, water samples were passed through a 0.2 µm syringe filter (Whatman GD/X) and absorbance of CDOM was measured on either a Cary Varian 50 spectrophotometer using a 10 cm quartz cell or an UltraPath liquid waveguide spectrometer with 2 m path length." Samples for the UltraPath measurements were not diluted. While absorbance can be saturated in the short part of the spectrum when using 2 m path length, this issue was not observed at 440 nm. In the visible part of the spectrum, the results are better in terms of precision than using a classical method with a 10 cm cuvette.

**Reviewer 2 Comment 17:** The reviewer notes that for Sect. 2.5 the categorization of optical variability across water bodies is interesting, and the details are certainly relevant. The reviewer also notes that the categorizations are rather subjective, and effectively used to explain outliers in the algorithm relationship. In the present state of the manuscript, the reviewer believes this seems quite subjective. The reviewer also believes for a universal algorithm, it seems highly relevant to look for underlying means for deviations in the relationship that would aid in how the algorithm is applied. The reviewer goes on to state that effectively, the authors have categorized the environment that results in deviating optical properties that do not perform well within the algorithm; however, the authors haven't utilized the dataset to hypothesize on specific, only general, mechanisms (e.g., minerogenic content of particles and refractive index, spectrally different CDOM, dominance of phytoplankton absorption and scattering on Kd relationships rather than a relatively generic "sediment resuspension"). The author admits that while quite difficult, the level of detail for the other sections seems to warrant this consideration. The reviewer concludes that the the authors have attempted to bypass this variability by using the end-member approach, targeting the wavelengths most and least influenced by aCDOM; other optical parameters significantly impacting the signal suggests a more detailed explanation, outside of categories, is warranted.

**Authors Response 37:** The authors have addressed the subjective classification topic in Authors Response 12 among others, and have established revisions in order to clarify this misunderstanding.

As presented in Sect. 4, the most extensive application of the optically complex data to the algorithm results in the use of 90% or more of all the data with an accuracy of approximately 6%. It is not scientifically objective to characterize this capability as resulting in "deviating optical properties that do not perform well within the algorithm." In fact, as noted in Sect. 4, standard algorithms, some of which do not span three decades of dynamic range in optical and biogeochemical parameters, do not perform as well (P31 L9–12), and these published algorithms do not provide the specificity requested by the reviewer to explain their large inaccuracies (some of which are significantly larger).

The authors do not believe that an algorithm that spans three decades of dynamic range in both optical and biogeochemical parameters and that has an accuracy of approximately 6% when 90% or more of the optically complex data are used needs to investigate the specific mechanisms for such a small degradation in accuracy; especially when the application of that same algorithm to conservative water masses has an accuracy of approximately 1% and the uncertainty in the optical measurements is on the order of 5%. In other words, the algorithm is clearly robust and significantly more capable than any present alternative.

The robustness is further established by creating a so-called universal algorithm, which is undefined by the reviewer, but is assumed to mean that any water mass wherein an optical profiler can be deployed is expected to be part of the evaluation of the end-member approach. In this case, the universal algorithm is constructed from *all the data from all subcategories*. The linear fit of this universal data set is $y = 0.2206x + 0.0088$, the RMSE is 7.5%, and the new slope is to within 13.7% of the original value presented by Hooker et al. (2013). In other words, the universal algorithm includes water masses that would not normally be included in an algorithm—i.e.. hypersaline, alkaline, and polluted lakes—and it equals or exceeds the performance of common so-called global algorithms.

If the hypersaline, alkaline, and polluted lakes are removed from the universal algorithm, the linear fit of this sixth comprehensive data set (the fifth comprehensive data set is presented in Authors Response 12) is $y = 0.2250x + 0.0024$, the RMSE is 6.8%, and the new slope is to within 12.0% of the original value presented by Hooker et al. (2013). The robustness of the universal and sixth comprehensive data sets can be evaluated by comparing the results to the fifth comprehensive algorithm, which used data from all the $N_1$–$N_4$ classes and completely fulfilled the dynamic range in $a_{\mathrm{CDOM}}(440)$, as presented in Authors Response 12. The linear fit of the fifth comprehensive data set is $y = 0.2317x - 0.0053$, the RMSE is 5.3%, and the new slope is to within 9.3% of the original value presented by Hooker et al. (2013).

Although of general scientific interest, the level of accuracy achieved with the universal, fourth, fifth, or sixth comprehensive data sets does not warrant investigations into the influence of minerogenic content of particles and refractive index, spectrally different CDOM, dominance of phytoplankton absorption and scattering on $K_d$ relationships or any other source of variance, because the accuracies of the universal and comprehensive algorithms do not warrant such investigations which are otherwise beyond the scope of the work described here.

Sect. 4 of the manuscript will be modified to include the algorithm performance results for the fifth,

sixth, and universal data sets; the new material will be presented after the fourth data set results in P30 L1.

**Reviewer 2 Comment 18:** The reviewer cites P16 L6-7 regarding new acreage from overfilled likes is a source of atypical constituents, either in composition or concentration, and states, "Really, this is the challenge of creating flexible, accurate algorithms that work across a variety of water types, either due to spatial, temporal or extreme event variability." The reviewer posits that it seems rather than addressing how to accurately estimate CDOM by modifying the algorithm, the authors highlight what is "abnormal" about these environments. The reviewer believes there is certainly room for this, but thinks the authors would be better served by focusing on how their algorithm could be adapted for these environments, rather than subjectively classifying environments where the algorithm does not perform well. The reviewer asserts the authors emphasize how the sensitivity of the instruments used detects these changes, but there is no analysis for how this increased sensitivity can be used to develop more capable algorithms.

**Authors Response 38:** Within the manuscript plus the Authors Responses herein, the $\Lambda_{780}^{320}$ algorithm is unchanged. What is allowed to change is the optical complexity of the data used to evaluate the algorithm. Furthermore, many of the water masses that are part of the optical complexity in the subcategories are properly labeled considered abnormal, e.g., hypersaline, alkaline, or polluted lakes. They were sampled with the purpose of providing extreme data to quantify how the performance of the algorithm is degraded by such water masses. The results presented in Authors Response 37 for the universal, fifth, and sixth data sets establish that the algorithm is sufficiently robust to provide accurate results even in the presence of such water masses and confirm a more capable algorithm is not needed.

In regards to the sensitivity argument, the sensitivity of the C-OPS and C-PrOPS instrumentation becomes critical in providing confidence in the derived relationships. Using other instrumentation (e.g., the Satlantic HyperPro II), demonstrates this, in that the curated NOMAD dataset includes what the manuscript demonstrates to be aberrant measurements. This is not obvious when using instruments with lower sensitivity, because the variance is large enough that it is not clear whether these are true outliers. The authors think this point is adequately expressed in the manuscript without explicitly calling out the shortcomings of specific instruments, which would be necessary to adequately discuss the per-instrument sensitivity (performance) of different datasets. Consequently, the manuscript will not be modified further.

**Reviewer 2 Comment 19:** The reviewer cites P18 L27–28 regarding how local wind conditions could elevate the values associated with a typical bloom into atypical concentrations and asserts this calls into question the purpose of classifications. The reviewer also notes these are conditions that will be observed, either through *in situ* or satellite observations and wonders if the algorithm could be improved by factoring in wind conditions.

**Authors Response 39:** The authors believe this is a good example as to why the subjective classification process is powerful, because it includes an external important forcing mechanism. For example, if the data from the observations involved in the local wind conditions cited above are used to evaluate the performance of the OC3M6 algorithm, the HPLC TChl $a$ concentration on the sheltered upwind side of a wind-blown polluted lake is $67.484 \, \text{mg} \, \text{m}^{-3}$ and $1,116.512 \, \text{mg} \, \text{m}^{-3}$ on the opposite downwind shore. The uncertainty in the OC3M6 algorithm for the lee side is $91.2\%$ and for the opposite shore it is $1,555.0\%$.

The authors are unaware of any example where local wind conditions were used to improve chlorophyll retrievals by factoring in wind conditions, but if the data from the polluted wind-blown lake were used to validate a chlorophyll algorithm, it is anticipated that a notation regarding anomalous concentrations from wind effects would likely be appreciated. Without an effort associated with wind-blown corrections, the algorithm presented in the manuscript for the wind-blown polluted lake for $a_{\text{CDOM}}(440)$ has an uncertainty on the lee side of the polluted lake of $24.5\%$ and $36.4\%$ on the opposite shore. These numbers are not so large as to render the $a_{\text{CDOM}}(440)$ algorithm useless, but this does occur for the OC3M6 algorithm. Consequently, no modifications will be made to the manuscript.

**Reviewer 2 Comment 20:** The reviewer cites P20 L22-24 in regards the phenomenon that as end members are brought spectrally closer together, the range of expression available to distinguish two similar but optically different water masses decreases and suggests the mechanics of this could be explored and explained.

**Authors Response 40:** This phenomenon is easily understood by studying Fig. 8, wherein the range in the optical axis for the $K_d(313)/K_d(875)$ algorithm is greater than the range in the optical axis for the $K_d(412)/K_d(670)$ algorithm. The nuances of legacy algorithms is not a principal focus of the study and the material presented in the manuscript is deemed sufficient, so the manuscript is not modified.

**Reviewer 2 Comment 21:** The reviewer cites P21 L16-17 "Application of $\Lambda_{670}^{412}$ data to the corresponding algorithm in Fig. 8 results in 13 observations with negative (predicted) $a_{\mathrm{CDOM}}(440)$ values, which are removed to leave 212 unique stations. This process demonstrates how end-member algorithms can be used to quality assure optical data in archives (Sect. 3.7)." The reviewer also cites P23 L2-4 "With respect to the algorithm, the increased bias, variance, and 13 negative derived values obtained with NOMAD data (which is a small, quality controlled subset of the larger NASA SeaBASS archive) in clearer waters suggests the legacy data are degraded by sampling artifacts." The reviewer then questions if this is an issue with the algorithm or the measurements.

**Authors Response 41:** The preponderance of evidence suggests that it is an issue with the legacy instrumentation. For the same geographical region under similar conditions, the C-OPS clear and turbid water data conform to the algorithm. The independent validation with the NOMAD dataset shows that the slope of the algorithm fit for the turbid partition of the NOMAD dataset is consistent with the slope found in the global algorithm perspective, but that the slope of the algorithm fit for the clear partition of the NOMAD dataset is significantly different (P22 L25–29).

Classification of the water mass can help assess whether the NOMAD issue arises from the algorithm or from the measurements, because the manuscript shows that algorithm performance varies with class assignment. Because the metadata for subjective classification does not exist for NOMAD, the objective classification scheme was applied to NOMAD and found the number of data in each class as follows: $N_1$ 6, $N_2$ 13, $N_3$ 135, $N_4$ 49, $N_5$ 0, and 9 were unclassified. All of the turbid data were in classes $N_3$ and $N_4$, but the clear data were included classes $N_1$ and $N_2$ plus the 9 observations that were not classified. This means the slope of the clear partition was determined with 19 points that were classified and 9 that were not, which accounts for the poor performance with respect to the algorithm. It also suggests that the measurements were the issue with the NOMAD data, because the spectra could not be classified.

This example of using objective classification as an analytical or investigative tool will be briefly summarized and added to the manuscript at the very end of Sect. 3.7 (P27 L13).

**Reviewer 2 Comment 22:** The reviewer cites P23 L14-15, "The data set established herein has an extensive number of observations directly suitable for validation exercises (Figs. 3 and 9) plus 15 subcategories (Sect. 2.5) of potentially (but not automatically) problematic water bodies (Figs. 4–7), with the latter determined subjectively." The reviewer asserts the authors acknowledge that "problematic" water bodies were determined subjectively, and these observations do not agree well with the algorithm while questioning a) if these observations span natural environmental variability that can be observed, and b) outside of directly observing human structures (e.g., reflectance of a shipwreck visible from surface waters), why did the authors not consider how to retrieve valid $a_{\mathrm{CDOM}}(440)$ values for these waters, and rather chose to assume the algorithm works very well and these waters are problematic? The reviewer also notes it isn't clear how the algorithm can be used to determine whether legacy data are valid or not, because its performance is based on these subjective classifications.

**Authors Response 42:** The authors do not acknowledge that problematic waters were determined subjectively. As stated in the manuscript and by the reviewer, subcategories were considered to be potentially, but not automatically, problematic. Also, the authors do not acknowledge that the subcategory data do not agree well with the algorithm, because this phrase does not appear in the manuscript. Furthermore, and as presented in Sect. 4 (plus Authors Response 12 and 23), data from the subcategories were incrementally added to assess algorithm performance and all of the algorithm assessments do not involve any description wherein performance is described as being degraded by observations that do not agree well with the algorithm, including new assessments presented in Authors Response 37.

All optical profiling was to the depth of the 10% or 1% light level while remaining above the bottom depth, so no data were contaminated by bottom or manmade structures. All the optical data retrieved valid $a_{\mathrm{CDOM}}(440)$ values using standard processing, except the UV domain for White Lake required channel-by-channel processing and are considered estimated, which is why they were omitted from the fourth, but

nonetheless comprehensive data set used as a comprehensive evaluation of the algorithm. The manuscript contains no statement that the algorithm works very well; all algorithm evaluations are provided in terms of quantified performance.

The ability of the algorithm to determine whether legacy data are valid was demonstrated in the manuscript with additional refinements in Authors Response 41. The argument that algorithm performance is based on subjective classifications requires the understanding that the non-conservative water masses are likely outside the range in the gradient of a constituent. As stated in the Introduction, the natural range of variability in water masses that may be exceeded by extreme events (but this does not imply that those events are not "natural," just that they are statistically anomalous). An objective classification scheme would still identify these data as "anomalous," but without ancillary (non-optical) data, it may not be obvious whether the anomalies are due to unusual water properties or lack of adherence to the protocols for measuring optical properties (see Authors Response 41).

The authors therefore suggest that if the optical data do not adhere to the algorithm, the first step would be to determine whether there are other factors (subjectively described within the 15 subcategories outlined in this manuscript). If there are no discernible reasons for the data to appear "anomalous" then it strongly suggests that there is an issue with data collection (see Authors Response 41), or that some environmental stressor not captured in the 15 subcategories resulted in an anomalous situation. In other words, the authors are not suggesting that strict adherence to the algorithm is a criteria by itself, but rather deviation from the algorithm should trigger additional scrutiny of the full dataset.

Clarification on these points will be improved through an addition to manuscript Sect. 4 (in P31 after L16) as described in Author Response 43.

**Reviewer 2 Comment 23:** The reviewer cites P24 L7-8, "Consequently, a subcategorization scheme based on the optical measurements alone might be advantageous to the validation process, particularly for archival data." This is assuming that CDOM should behave conservatively across water masses? It seems the very point the paper is making is that abnormal environments produce CDOM of a different spectral nature. This is important. Why have the authors not attempted to accurately estimate this variability?.

**Authors Response 43:** One point of the study is that for the vast majority of water bodies (by surface area), a single global algorithm effectively retrieves CDOM. For a subset of anomalous cases, there are two potential explanations for deviations (anomalies) from the expected fit: a) there is an unusual environmental factor occurring (and 15 subcategories are provided for evaluation, most of which do not significantly degrade the global algorithm), or b) the data were collected improperly (either the diffuse attenuation coefficients or the CDOM absorption coefficient values). Application of the algorithm to historical (archival) data will identify outliers, and this would help guide a more careful analysis of the preponderance of evidence for putting those data into one or the other category (i.e., the water mass is truly anomalous or the data collection is suspect).

To address the points in this comment, as well as for Reviewer 2 Comment 22, the manuscript will be modified in P31 after L16 by adding the following text: "Screening of newly collected or archival data with respect to a selected algorithm can be accomplished by initially flagging data points more than 12% from the expected relationship, and then more carefully examining those points using both objective and subjective criteria (based on available metadata) to determine whether the results are expected, or are more likely to indicate a problem with data collection procedures."

**Reviewer 2 Comment 24:** The reviewer cites P23 L18-24, "The author's reference clustering analysis, particularly fuzzy clustering presented by Moore et al. (various years)" and questions why subjective categorizations were used rather than an objective approach such as that of Moore et al.? The reviewer also suggests it would also be useful to reference this work earlier, perhaps in the introduction, and discussing the need for classifications to build effective algorithms could also be elaborated.

**Authors Response 44:** To provide insights into the dynamics of the observed $K_d$ spectra, all data were first categorized into subcategories based on a subjective classification but using sampling information necessary to better understand optical properties, which would otherwise not be known. The robustness of the subcategories are logically supported by an objective classification using fuzzy c-means (FCM). Please note that FCM is different from a crisp or hard classification such as $k$-means. The authors believe that these two

steps are required to appropriately interpret the results. Following the reviewer, a brief description about necessity of both subjective and objective classifications was added to Sect. 1 (P4 L18–25).

**Reviewer 2 Comment 25:** The reviewer cites P26 L11-14, "The decrease in the percent composition of the validation quality data as a function of increasing class number ($N_1$–$N_5$) is an indicator of the difficulty of validating an algorithm within increasingly complex waters. The recurring contribution of a relatively small number of principal subjective subcategories to the gradient in optical complexity starting with $N_2$ and then continuing for $N_3$–$N_5$ confirms the original subcategory approach has merit." The reviewer then posits that conversely, the end member approach only performs well in bodies of water with little optical variability. The reviewer states it isn't clear why a subjective deconstruction of water bodies was used versus a fuzzy clustering approach where alternate relationships between $K_d$ and $a_{\mathrm{CDOM}}(440)$ were explored, noting that fuzzy clustering approaches use an objective classification scheme to separate out waters with the intention of providing a framework where different algorithms can be applied, and questions why was that not explored.

**Authors Response 45:** One of the objectives of the study was to examine a global algorithm for estimating $a_{\mathrm{CDOM}}(440)$ that works for a diversity of water masses, which is a different approach compared to what the reviewer mentioned, e.g., Hieronymi et al. (2017). While the authors explained "anomalies" using both subjective and objective classifications, overall performance of the algorithm was always superior to or in keeping with global algorithms in use by the community of practice—even when the present study incrementally added in increasingly complex water masses beyond the capabilities of existing global algorithms (see Authors Response 12, 23, and 37). Consequently, no additional modifications were made to the manuscript.

**Reviewer 2 Comment 26:** The reviewer cites P26 L28 and the use of "evolving conservatively" and then questions a) if this is being used to represent anything that is a natural process within the water column, with no added optical constituents, and b) whether photo- and microbial degradation of CDOM considered a conservative process. The reviewer suggests a clearer discussion of Case 1 and Case 2 waters and how that classification relates to the classification used here would clarify this.

**Authors Response 46:** The manuscript already answers the two questions on P3 L28 to P4 L3 wherein evolution that is within a constrained (natural) range of a water mass are considered conservative. The requested clarifications regarding case-1 and case-2 waters already appear in the manuscript on P25 L18 to P26 L1, but additional small modifications will be added to P26 L2–10 to improve clarity and provide completeness.

Respectfully yours,

Stanford Hooker